# Robust Conformal Prediction under Joint Distribution Shift

## Abstract

Uncertainty prevails due to the lack of knowledge about data or model, and conformal prediction (CP) predicts multiple potential targets, hoping to cover the true target with a high probability. Regarding CP robustness, importance weighting can address covariate shifts, but CP under joint distribution shifts remains more challenging. Prior attempts addressing joint shift via $f$-divergence ignores the nuance of calibration and test distributions that are critical for coverage guarantees. More generally, with multiple test distributions shifted from the calibration distribution, simultaneous coverage guarantees for *all* test domains requires a new paradigm. We design *Multi-domain Robust Conformal Prediction (mRCP)* that first formulates the coverage difference that importance weighting fails to capture under any joint shift. To squeeze such coverage difference and guarantee the $(1 - \alpha)$ coverage in all test domains, we propose *Normalized Truncated Wasserstein distance (NTW)* to comprehensively capture the nuance of any test and calibration conformal score distributions, and design an end-to-end training algorithm incorporating NTW to provide elasticity for simultaneous coverage guarantee over distinct test domains. With diverse tasks (seven datasets) and architectures (black-box and physics-informed models), NTW strongly correlates (Pearson coefficient=0.905) with coverage differences beyond covariate shifts, while mRCP reduces coverage gap by $50\%$ on average robustly over multiple distinct test domains.

## 1 Introduction

The growing data volume, enhanced computation capability, and advanced models significantly improve machine learning predictive accuracy. Nevertheless, noises, unobservable factors, and the lack of knowledge lead to uncertainty that stakeholders should ponder along model predictions when making decisions particularly in areas such as fintech [25], autonomous driving [2], traffic forecasting [4], and epidemiology [32, 27]. Conformal Prediction (CP) addresses uncertainty by predicting a set of possible target(s) rather than a single guess [31]. Specifically, CP computes conformal scores (residuals between predicted and true targets for regression tasks) of a trained model $f$ on a calibration set, and calculates the $1 - \alpha$ quantile $q$ of these scores. For any input $x$, CP produces the smallest prediction set $C(x)$ consisting of target values whose conformal scores are less than $q$. Assuming that the test and calibration data are exchangeable (including i.i.d.), the true target $y$ is guaranteed to be covered by $C(x)$ with at least $1 - \alpha$ probability.

In practice, calibration distribution $P_{XY}$ and test distribution $Q_{XY}$ may differ thus $P_{XY} \neq Q_{XY}$, termed as **joint distribution shift** and violate the exchangeability assumption. Joint shift can occur with either covariate shift ($P_X \neq Q_X$) or concept shift ($P_{Y|X} \neq Q_{Y|X}$), though what causes a joint shift is difficult to infer from the observed data only. With importance weighting, covariate shift is shown not to affect the coverage confidence guarantee [29]. To address CP under joint

Submitted to 38th Conference on Neural Information Processing Systems (NeurIPS 2024). Do not distribute.

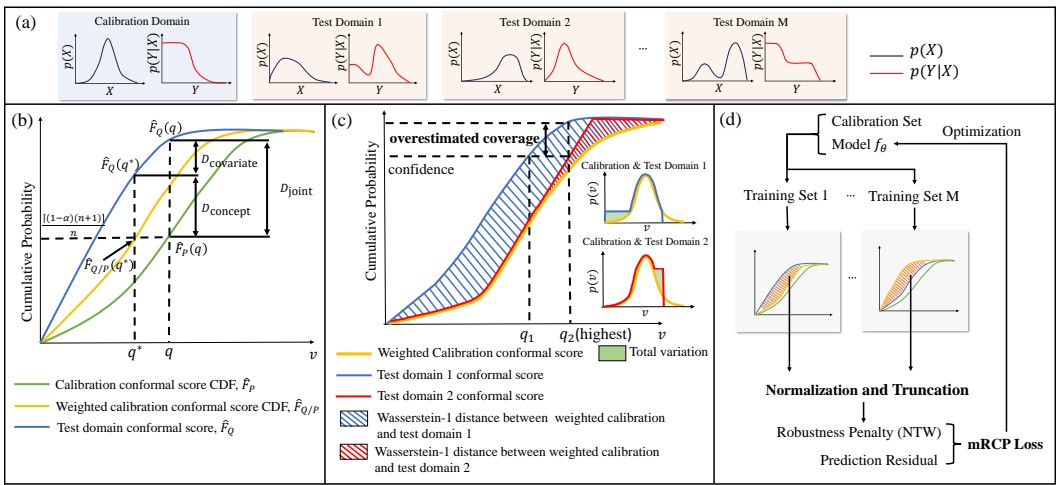

Figure 1: (a) Multiple test domains $\mathcal{E} = \{e_1, ..., e_M\}$ with joint shifts ($Q_{XY}^{(e)} \neq P_{XY}$); (b) coverage difference $D_{\text{joint}} = \hat{F}_Q(q) - \hat{F}_P(q)$ (Eq. (5)) due to $Q_{XY}^{(e)} \neq P_{XY}$ is decomposed into the $D_{\text{covariate}}$ (Eq. (7)) caused by covariate shift ($Q_X^{(e)} \neq P_X$) and the remaining $D_{\text{concept}}$ (Eq. (8)) due to concept shift ($Q_{Y|X}^{(e)} \neq P_{Y|X}$); (c) Wasserstein-1 (W-) distances (Eq. (11)) between test and weighted calibration conformal score CDFs capture the expected $D_{\text{concept}}$ (Eq. (10)). However, $f$-divergence (e.g., total variation, KL divergence) does not compare two CDFs pointwisely and fails to capture such an expectation. Test domains 1 and 2 both have identical total variations to the calibration domain but different W-distances. With multiple test domains, using a single $1 - \alpha$ quantile [33, 6] lead to the dilemma of CP coverage efficiency and confidence guarantee; (d) Solution: Normalized Truncated W-distance (Eq. (16)) is robust to outlier scores and different score scales across test domains, and the mRCP algorithm reduces NTW on all test domains can elastically train a model to guarantee conformal coverage for $Q_{Y|X}^{(e)} \neq P_{Y|X} \forall e \in \mathcal{E} = \{e_1, ..., e_M\}$.

shift, $f$-divergence is adopted in [33, 6] to measure the difference between $P_{XY}$ and $Q_{XY}$ or the corresponding conformal scores distributions. However, $f$-divergence ignores where the two distributions differ, which quantiles and coverage guarantees depend on (Figure 1, (c)). When test data are sampled from multiple distinct test distributions $Q_{XY}^{(e)}, e \in \mathcal{E} = \{e_1, ..., e_M\}$, it is desired to ensure simultaneous $1 - \alpha$ coverage for all test distributions. Previous work selects the highest $1 - \alpha$ quantile from all test distributions and constructs $C(x)$ for $x \in Q_X^{(e)}, \forall e \in \mathcal{E} = \{e_1, ..., e_M\}$, producing excessively large set $C(x)$. Selecting other quantiles may lead to smaller coverage on a test domain than was expected during calibration, leading to prediction overconfidence. Without a new paradigm to guarantee coverage under multiple shifted test distributions, the dilemma between CP coverage efficiency and confidence guarantee seems unavoidable.

We first decompose the coverage difference under any joint distribution shift to a component due to covariate shift ($P_X \neq Q_X^{(e)}$, addressed by importance weighting [29]) and that due to concept shift ($P_{Y|X} \neq Q_{Y|X}^{(e)}$). We propose *Normalized Truncated Wasserstein distance (NTW)* to robustly capture where the test and importance-weighted calibration conformal score cumulative density function (CDF) deviate (Figure 1, (b)). We design *Multi-domain Robust Conformal Prediction (mRCP)* by minimizing all NTW terms over $\mathcal{E} = \{e_1, ..., e_M\}$ during model training (Figure 1, (d)) to elastically guarantee coverage confidence for all test domains. Experiments on regression tasks on seven datasets demonstrate that: *1)* NTW well-correlates with the coverage difference after importance weighting (Pearson coefficient 0.905); *2)* mRCP provides conformal predictions that reduce average coverage difference by $50\%$ compared to baselines under multiple joint shifts; *3)* mRCP is sufficiently general to address joint distribution shifts even after incorporating domain knowledge when available.

## 2 Background and related work

### 2.1 Conformal prediction

Let $x \in \mathcal{X}$ and $y \in \mathcal{Y}$ denote the input and output random variable, respectively, where $\mathcal{X}$ and $\mathcal{Y} \subseteq \mathbb{R}$ is the input and output space, respectively. On $\mathcal{X} \times \mathcal{Y}$, the calibration domain is defined by a

joint distribution $P_{XY}$, and we consider a calibration set $S_c = \{(x_1, y_1), \ldots, (x_n, y_n)\}$ are drawn *i.i.d.* from $P_{XY}$. Similarly, a test set $S_t = \{(x_1, y_1), \ldots, (x_m, y_m)\}$ is drawn *i.i.d.* from test domain, which is defined by a joint distribution $Q_{XY}$.

With a trained regression model $f$, the conformal score $v_i = v(x_i, y_i) = |f(x_i) - y_i|$ is the residual between the predicted target $f(x_i)$ and the true target $y_i$. The set of calibration conformal scores is denoted as $V_c = \{v(x_i, y_i)|(x_i, y_i) \in S_c\}$. Let $q$ be the $\lceil(1 - \alpha)(n + 1)\rceil/n$ quantile of $V_c$:

$$q = \text{Quantile}\left(\frac{\lceil(1 - \alpha)(n + 1)\rceil}{n}, \frac{1}{n}\sum\nolimits_{v_i \in V_c} \delta_{v_i}\right), \tag{1}$$

where $\delta_{v_i}$ represents the point mass at $v_i$ (i.e., the distribution placing all mass at the value $v_i$). Quantile$(1 - \alpha, F) := \inf\{z|\text{Pr}(Z < z) \geq 1 - \alpha\}$ and $F$ is the CDF of $Z$. With the quantile $q$, the CP prediction set of an input $x$ from $S_t$ is

$$C(x) = \{\hat{y} \in \mathbb{R}||f(x) - \hat{y}| \leq q, (x, y) \in S_t\}. \tag{2}$$

Most CP methods, such as[22, 23], rely on the assumption of exchangeability, which is relaxed from the i.i.d. assumption [31]. In our scenario, if the calibration and test samples are drawn from the identical joint probability distribution ($P_{XY} = Q_{XY}$), these calibration and test samples are i.i.d. Under this assumption, the probability that the true target $y$ is included in $C(x)$ is at least $1 - \alpha$, which is called **coverage guanrantee**, or more formally,

$$\text{Pr}\left(y \in C(x)\right) \geq 1 - \alpha. \tag{3}$$

## 2.2 Conformal prediction under domain shift

**Covariate shift** ($P_X \neq Q_X$) means marginal distributions between the calibration and test domains are different. CP under covariate shift is addressed using importance weighting [29]. Under a probabilistic view, [14] defined the covariate shift as a bounded perturbation on any test input and developed adaptive probabilistically robust CP. The condition of multiple test domains is discussed in [15], and similar topics include coverages under feature-stratification [7, 11].

**Joint distribution shift** ($P_{XY} \neq Q_{XY}$) indicates at least one of covariate shift ($P_X \neq Q_X$) and concept shift (different conditional distributions, $P_{Y|X} \neq Q_{Y|X}$) will occur [17]. This shift is more general and the importance weighting method cannot address changes in conditional distribution. With $M$ test domains $\mathcal{E} = \{e_1, ..., e_M\}$, each $e \in \mathcal{E}$ is defined by a joint distribution $Q_{XY}^{(e)}$ and holds a joint shift with calibration domain $P_{XY}$ (i.e., $P_{XY} \neq Q_{XY}^{(e)}$). Considering this condition, previous works, such as [6, 33], presume all test domains fall in a predefined $f$-divergence range, calculate confidence-specified quantile of each test domain, and apply the highest quantile to all domains. This method causes excessively high coverages and thus overlarge prediction sets, which reduces prediction efficiency because smaller prediction sets can help locate true targets better.

# 3 Conformal prediction under joint distribution shift

## 3.1 Decomposition of coverage difference

We decompose the coverage difference between a calibration domain $P_{XY}$ and a test domain $Q_{XY}$ under **joint distribution shift** at a user-specified confidence $(1 - \alpha)$.

Similar to $V_c$, we define the test conformal score set $V_t = \{v(x_i, y_i)|(x_i, y_i) \in S_t\}$. With the indicator function $\mathbb{1}$, empirical CDFs of calibration and test conformal scores are

$$\hat{F}_P(v) = \frac{1}{n}\sum\nolimits_{v_i \in V_c} \delta_v \mathbb{1}_{v_i < v}, \quad \hat{F}_Q(v) = \frac{1}{m}\sum\nolimits_{v_i \in V_t} \delta_v \mathbb{1}_{v_i < v}. \tag{4}$$

With given $1 - \alpha$ confidence, quantile $q$ is calculated in Eq. (1), and the coverage difference under a joint distribution shift can be quantified as

$$D_{\text{joint}}(q) = \hat{F}_Q(q) - \hat{F}_P(q). \tag{5}$$

[29] employs importance weighting for CP under covariate shift. Specifically, if the ratio of test to calibration covariate likelihoods, $Q_X/P_X$, is known, a calibration conformal score $v_i \in V_c$ is

weighted by $p_i = w(x_i)/\sum_{j=1}^n w(x_j)$, where $w(x_i) = Q_X(x_i)/P_X(x_i)$. Therefore, the empirical CDF of weighted empirical calibration scores is given by

$$\hat{F}_{Q/P}(v) = \sum_{i=1}^n p_i \delta_{v_i} \mathbb{1}_{v_i < v},$$

where the subscript $Q/P$ indicates conformal scores of calibration domain $P$ is weighted by conformal scores of test domain $Q$. The confidence-specified quantile of the weighted calibration conformal scores is

$$q^* = \text{Quantile}\left(\lceil (1-\alpha)(n+1)\rceil/n, \sum_{i=1}^n p_i \delta_{v_i}\right). \tag{6}$$

As importance weighting ensures the $1 - \alpha$ coverage as though covariate shift were absent, coverage difference $D_{\text{covariate}}$ caused by covariate shift is the gap between the coverages under test conformal score CDF using quantiles on unweighted and weighted calibration conformal score distributions.

$$D_{\text{covariate}}(q, q^*) = \hat{F}_Q(q) - \hat{F}_Q(q^*). \tag{7}$$

Importance weighting can not address CP under joint shift as it fails to capture changes in conditional probability distribution caused by concept shift, thus we present the coverage difference caused by concept shift is

$$D_{\text{concept}}(q, q^*) = D_{\text{joint}}(q) - D_{\text{covariate}}(q, q^*) = \hat{F}_Q(q^*) - \hat{F}_P(q), \tag{8}$$

which is remaining coverage difference after applying importance weighting. Here we assume $\hat{F}_P(q) = \hat{F}_{Q/P}(q^*)$, so we can rewrite $D_{\text{concept}}$ by

$$D_{\text{concept}}(q^*) = \hat{F}_Q(q^*) - \hat{F}_{Q/P}(q^*). \tag{9}$$

The error bound for the assumption is quite small especially when the calibration set size $n$ is large. The detailed proof is provided in Appendix B. We denote $D_{\text{concept}}$ as $D$ for simplification.

## 3.2 Normalized Truncated Wasserstein distance

To develop a metric that is independent of confidence level and can quantify the overall closeness between weight calibration and test conformal scores, we estimate the expected coverage difference under concept shift as

$$\mathbb{E}[D] = \frac{1}{n} \sum_{v_i \in V_c} \left| \hat{F}_Q(v_i) - \hat{F}_{Q/P}(v_i) \right|, \tag{10}$$

based on the approximation in Eq. (9), where $\mathbb{E}$ indicates the expectation function.

**Definition 1** (Wasserstein-1 Distance). *If $F_1$ and $F_2$ are two cumulative distribution functions (CDFs), the Wasserstein-1 distance, $d_W$, is quantified by the area between $F_1$ and $F_2$.*

$$d_W(F_1, F_2) = \int_{\mathbb{R}} |F_1(v) - F_2(v)| dx. \tag{11}$$

Applying Wasserstein-1 distance (W-distance) in Eq. (11) to $\hat{F}_Q$ and $\hat{F}_{Q/P}$, we get

$$d_W(\hat{F}_Q, \hat{F}_{Q/P}) = \int_0^\infty |\hat{F}_Q(v) - \hat{F}_{Q/P}(v)| dv. \tag{12}$$

As we define conformal scores as the residuals between predicted and true targets, they are always positive, so we only need to integral from 0 to $\infty$ in Eq. (12).

We assume $V_c$ is sorted. As both $\hat{F}_Q$ and $\hat{F}_{Q/P}$ are empirical CDFs, we can approximately represent $d_W(\hat{F}_Q, \hat{F}_{Q/P})$ in a discrete form as

$$d_W(\hat{F}_Q, \hat{F}_{Q/P}) \approx \sum_{i=1}^{n-1} \left| \hat{F}_Q(v_i) - \hat{F}_{Q/P}(v_i) \right| (v_{i+1} - v_i), \quad v_i \in V_c. \tag{13}$$

Eq. (13) shows $d_W(\hat{F}_Q, \hat{F}_{Q/P})$ can be estimated as a weighted summation of $|\hat{F}_Q(v_i) - \hat{F}_{Q/P}(v_i)|$ for $v_i \in V_c \backslash \{v_n\}$ with the corresponding weight $v_{i+1} - v_i$. Also, Eq. (10) indicates that $\mathbb{E}[D]$ can be regarded as the weighted summation of $|\hat{F}_Q(v_i) - \hat{F}_{Q/P}(v_i)|$ for $v_i \in V_c$ with weight $1/n$. The

similarity between Eq. (13) and Eq. (10) allows us to apply the W-distance between the test and weighted calibration conformal score to capture expected coverage difference under concept shift.

Care needs to be taken for Eq. (13) to make this metric more robust. At first, we expect the weights $v_{i+1} - v_i$ to be approximately equal, as weights in Eq. (10) are constants $1/n$. However, some outlier calibration conformal scores have large distances from their neighbors, causing involved weights much higher than $1/n$. These outlier scores are represented as a long tail of $\hat{F}_{Q/P}$ when it converges to 1. Therefore, it is necessary to establish a partition threshold to truncate the long tail. We calculate the partition threshold

$$v_\sigma = \inf \left\{ v_i | \hat{F}_{Q/P}(v_i) \geq 1 - \sigma, v_i \in V_c \right\}, \tag{14}$$

which is the smallest calibration conformal score whose coverage is greater or equal to a user-defined value $1 - \sigma$. In contrast to the original $d_{\mathrm{W}}(\hat{F}_Q, \hat{F}_{Q/P})$ integrated on the set of real numbers, the truncated form is integrated from 0 to $v_\sigma$ as

$$d_{\mathrm{TW}}(\hat{F}_Q, \hat{F}_{Q/P}) = \int_0^{v_\sigma} |\hat{F}_Q(v) - \hat{F}_{Q/P}(v)| dv. \tag{15}$$

Secondly, as the summation of weights in Eq. (10) is 1, we also need to divide each $v_{i+1} - v_i$ by $v_\sigma - v_1$. When the calibration set is large enough, it is plausible to assume the existence of a calibration sample fitting the trained model $f$ very well, causing the smallest calibration conformal score $v_1 \approx 0$. Therefore, this normalized can be formulated as

$$d_{\mathrm{NTW}}(\hat{F}_Q, \hat{F}_{Q/P}) = \frac{1}{v_\sigma} \int_0^{v_\sigma} |\hat{F}_Q(v) - \hat{F}_{Q/P}(v)| dv. \tag{16}$$

A lower $d_{\mathrm{NTW}}$ indicates more similarity between $\hat{F}_{Q/P}$ and $\hat{F}_Q$, thus leading to more robust conformal prediction in the test domain. As a result, NTW enables us to assess the expected coverage difference due to concept shift in Eq. (10). Experiment results in Section 5 and Appendix E show the necessity of truncation and normalization. We also prove that the W-distance between the test and weighted calibration conformal score population CDF can establish an upper bound for coverage difference under concept shift in Appendix C.

## 4 Multi-domain robust conformal prediction

If a calibration set $S_c$, and a test set $S_t$ are drawn from a domain $P_{XY}$, the i.i.d. assumption is satisfied, and the coverage guarantee in Eq. (3) holds for $(x, y) \in S_t$.

The domain $P_{XY}$ can be decomposed into $M$ multiple domains, denoted as $\mathcal{E} = \{e_1, ..., e_M\}$.

$$P_{XY}(x, y) = \frac{1}{M} \sum_{e \in \mathcal{E}} Q_{XY}^{(e)}(x, y) \tag{17}$$

However, for $e \in \mathcal{E}$, denote $S_t^{(e)}$ a test set drawn from $Q_{XY}^{(e)}$, then the coverage guarantee may no longer hold for $(x, y) \in S_t^{(e)}$, because joint distribution shift may occur between $P_{XY}$ and $Q_{XY}^{(e)}$. It indicates CP can be overconfident and underconfident for samples from different $Q_{XY}^{(e)}$, resulting in prediction biases.

Inspired by the works of multi-domain generalization [26, 18, 19, 1], we propose **Multi-domain Robust Conformal Prediction (mRCP)** to make the coverage approach confidence in all domains, using a training set $S^{(e)}$ from the data distribution $Q_{XY}^{(e)}$ for $e \in \mathcal{E}$ and a calibration set $S_c$ from $P_{XY}$.

The objective function of mRCP includes two components. First, for the minimization of prediction residuals, denoting $l$ a loss function, Empirical Risk Minimization (ERM) [30] is incorporated as

$$\mathcal{L}_{\mathrm{ERM}}(\theta) = \sum_{e \in \mathcal{E}} \mathcal{L}^{(e)}(\theta) = \sum_{e \in \mathcal{E}} \mathbb{E}_{(x_i, y_i) \sim S^{(e)}} [l(f_\theta(x_i), y_i)]. \tag{18}$$

Secondly, we aim for robust conformal prediction on each domain during testing, seeking a low value of $\mathbb{E}[D]$ in Eq. (10) across test domains, so mRCP needs to address coverage differences due to covariate and concept shifts simultaneously. To remove coverage differences due to covariate shifts,

168 it applies importance weighting to each domain $e \in \mathcal{E}$ during training and obtains $\hat{F}_{Q^{(e)}/P}$, which is
169 the calibration conformal score CDF weighted by $Q_{XY}^{(e)}$.

170 Besides, as we have a training set $S^{(e)}$ from domain $Q_{XY}^{(e)}$, an empirical CDF of conformal scores in
171 $Q_{XY}^{(e)}$ can be computed, denoted as $\hat{F}_{Q^{(e)}}^{tr}$. NTW quantifies the expected coverage difference caused
172 by concept shift between $\hat{F}_{Q^{(e)}/P}$ and training conformal score CDF $\hat{F}_{Q^{(e)}}^{tr}$. Combining these two
173 components, the objective function of mRCP is

$$\mathcal{L}_{\text{mRCP}}(\theta) = \sum_{e \in \mathcal{E}} \mathcal{L}^{(e)}(\theta) + \beta \sum_{e \in \mathcal{E}} d_{\text{NTW}}(\hat{F}_{Q^{(e)}}^{tr}, \hat{F}_{Q^{(e)}/P}), \tag{19}$$

where $\beta$ is a hyperparameter balancing these two parts. mRCP algorithm is shown in Algorithm 1.

---

**Algorithm 1** Multi-domain Robust Conformal Prediction

---

**Require:** $M$ training sets $S^{(e)}$, $e \in \mathcal{E}$; one calibration set $S_c$; $N$ training epochs; model $f_\theta$; partition value $\sigma$; loss function $l$; penalty hyperparameter $\beta$.
1: **for** $e \in \mathcal{E}$ **do**
2:      **for** $(x_i, y_i) \in S_c$ **do**
3:          $w(x_i) = \frac{Q_X^{(e)}(x_i)}{P_X(x_i)}, p_{(i,e)} = \frac{w(x_i)}{\sum_{j=1}^n w(x_j)}$       $\triangleright$ Covariate shift between $Q_{XY}^{(e)}$ and $P_{XY}$
4:      **end for**
5: **end for**
6:
7: **for** $i = 1$ to $N$ **do**
8:      $V_c = \{v(x_i, y_i) | (x_i, y_i) \in S_c\}$                $\triangleright$ Calibration score set
9:      **for** $e \in \mathcal{E}$ **do**
10:          $\mathcal{L}^{(e)}(\theta) = \mathbb{E}_{(x_i, y_i) \sim S^{(e)}} [l(f_\theta(x_i), y_i)]$       $\triangleright$ ERM loss of domain $e$
11:          $V^{(e)} = \left\{v(x_i, y_i) | (x_i, y_i) \in S^{(e)}\right\}$     $\triangleright$ Training score set of domain $e$
12:          $\hat{F}_{Q^{(e)}}^{tr} = \sum_{v_i \in V^{(e)}} \delta_{v_i} \mathbb{1}_{v_i \leq v}$        $\triangleright$ Training score CDF of domain $e$
13:          $\hat{F}_{Q^{(e)}/P}(v) = \sum_{v_i \in V_c} p_{(i,e)} \delta_{v_i} \mathbb{1}_{v_i \leq v}$     $\triangleright$ Calibration score CDF weighted by $Q_{XY}^{(e)}$
14:          $v_\sigma = \inf \left\{\hat{F}_{Q^{(e)}/P}(v_i) \geq 1 - \sigma, v_i \in V_c\right\}$     $\triangleright$ Truncation threshold
15:          $d_{\text{NTW}}\left(\hat{F}_{Q^{(e)}}^{tr}, \hat{F}_{Q^{(e)}/P}\right) = \frac{1}{v_\sigma} \int_0^{v_\sigma} \left|\hat{F}_{Q^{(e)}}^{tr}(v) - \hat{F}_{Q^{(e)}/P}\right| dv$     $\triangleright$ NTW calculation
16:      **end for**
17:      Optimize $f_\theta$ based on $\mathcal{L}_{\text{mRCP}}(\theta) = \sum_{e \in \mathcal{E}} \mathcal{L}^{(e)}(\theta) + \beta \sum_{e \in \mathcal{E}} d_{\text{NTW}}\left(\hat{F}_{Q^{(e)}}^{tr}, \hat{F}_{Q^{(e)}/P}\right)$
18: **end for**

---

174

# 5 Experiment

176 In this section, we validate NTW in Eq. (16) as a good indicator of expected coverage difference due
177 to concept shift and demonstrate the effectiveness of mRCP in obtaining coverage robustness across
178 different test domains.

## 5.1 Datasets and models

180 We conducted experiments across various datasets: (a) the airfoil self-noise dataset [5]; (b) Seattle-
181 loop [9], PeMSD4, PeMSD8 [16] for traffic speed prediction; (c) US-Regions, US-States, and
182 Japan-Prefectures [10] for epidemic spread forecasting. The airfoil dataset was manually altered to
183 create three subsets demonstrating covariate and concept shifts. 24 domains for the traffic datasets
184 were designated based on data generation hours, while epidemic dataset instances were categorized
185 into four domains reflecting different pandemic stages. A multilayer perceptron (MLP) with a (input
186 dimension, 64, 64, 1) architecture was utilized for all datasets. Traffic and epidemic prediction tasks
187 were also trained on corresponding physics-informed partial differential equations (PDEs), which are
188 the Susceptible-Infected-Recovered (SIR) model and the Reaction-Diffusion (RD) model respectively.
189 We refer to Appendix D for detailed experiment setups.

## 5.2 Experiments of NTW

For each of the experiment setups, a training set, a validation set, and a test set were sampled from each $Q_{XY}^{(e)}$ for $e \in \mathcal{E}$. One calibration set was sampled from $P_{XY}$ which is a mixture probability distribution of $Q_{XY}^{(e)}$ for $e \in \mathcal{E}$, as shown in Eq. (17). To validate NTW is a good indicator of $\mathbb{E}[D]$, we only need to use ERM in Eq. (18) to train the model $f_\theta$, which can be an MLP or a PDE. The loss function $l$ is the $\ell_1$ norm, as same as how we compute conformal scores.

After training, for $e \in \mathcal{E}$, we first calculated the NTW between the calibration conformal score CDF weighted by $Q_X^{(e)}/P_X$, and validation conformal score CDF of $Q_X^{(e)}$. Denote the NTW of domain $e$ as $d_{\mathrm{NTW}}^{(e)}$. Then, we estimated the expected coverage difference caused by concept shift on a test domain $e$, denoted as $\mathbb{E}_\alpha[D^{(e)}]$, using the coverage difference expectation between the test and weighted calibration conformal score CDFs on a $1 - \alpha$ confidence set $\{0.1, ..., 0.9\}$.

$\mathbb{E}_\alpha[D^{(e)}]$ and $d_{\mathrm{NTW}}^{(e)}$ should have a positive correlation for $e \in \mathcal{E}$, proving NTW can capture the expected coverage difference caused by concept shift.

**Baselines:** We select six baseline metrics to validate the effectiveness of NTW. Total variation $d_{\mathrm{TV}}$ [13], and Kullback-Leibler (KL) divergence $d_{\mathrm{KL}}$ [21] are chosen as two typical $f$-divergence metrics. Expectation difference $\Delta\mathbb{E}$ [19] is selected since it is a widely applied generalization metric. We also measure standard, normalized, and truncated W-distance, denoted as $d_{\mathrm{W}}$, $d_{\mathrm{NW}}$, and $d_{\mathrm{TW}}$ respectively, to demonstrate applying normalization and truncation together is necessary.

**Metric:** We apply the **Pearson coefficient** to quantify the correlations between metrics and the coverage difference expectation. It measures the linear correlation between two values by giving a value between -1 and 1 inclusive. 1,0, and -1 indicate perfect positive linear, no linear, and negative linear correlations, respectively. Therefore, if the Pearson coefficient of a metric is **higher**, this metric can indicate the expected coverage difference **better**. We provide a detailed definition of the Pearson coefficient in Appendix E.

Table 1: Pearson coefficients between metrics and coverage difference expectation under concept shift

| Dataset | Model | $d_{\mathrm{NTW}}$ | $d_{\mathrm{TV}}$ | $d_{\mathrm{KL}}$ | $\Delta\mathbb{E}$ | $d_{\mathrm{W}}$ | $d_{\mathrm{NW}}$ | $d_{\mathrm{TW}}$ |
|---|---|---|---|---|---|---|---|---|
| Airfoil | MLP | **1.000** | -0.356 | -0.545 | 0.891 | 0.878 | 0.951 | 0.967 |
| Seattle- | MLP | **0.971** | 0.461 | 0.054 | 0.781 | 0.759 | 0.762 | 0.765 |
| loop | PDE | **0.996** | 0.890 | 0.058 | 0.897 | 0.893 | 0.909 | 0.921 |
| PeMSD4 | MLP | **0.992** | 0.846 | -0.390 | 0.926 | 0.915 | 0.964 | 0.941 |
|  | PDE | **0.986** | 0.682 | -0.068 | 0.858 | 0.872 | 0.928 | 0.858 |
| PeMSD8 | MLP | **0.905** | 0.397 | -0.089 | 0.333 | 0.267 | 0.371 | 0.529 |
|  | PDE | **0.827** | 0.129 | -0.114 | 0.253 | 0.118 | 0.141 | 0.527 |
| US- | MLP | **0.999** | 0.966 | 0.965 | 0.872 | 0.885 | 0.912 | 0.931 |
| States | PDE | **0.999** | 0.966 | 0.964 | 0.817 | 0.848 | 0.890 | 0.899 |
| US- | MLP | **0.636** | -0.530 | -0.338 | -0.205 | -0.308 | -0.352 | -0.405 |
| Regions | PDE | **0.709** | 0.308 | 0.350 | 0.484 | 0.355 | 0.322 | 0.137 |
| Japan- | MLP | **0.996** | 0.986 | 0.988 | 0.943 | 0.948 | 0.954 | 0.950 |
| Prefectures | PDE | **0.997** | 0.983 | 0.981 | 0.907 | 0.918 | 0.935 | 0.924 |
| Average | | **0.905** | 0.574 | 0.325 | 0.619 | 0.583 | 0.607 | 0.629 |
| Standard Deviation | | **0.128** | 0.474 | 0.562 | 0.368 | 0.420 | 0.437 | 0.428 |

**Results:** Table 1 illustrates the Pearson coefficients between NTW and the coverage difference expectation among seven datasets and different models, compared with the other six baseline metrics. We highlight that NTW keeps holding the largest Pearson coefficient among all experiment setups, which means the proposed metric can keep indicating the coverage difference expectation. Specifically, the coefficients of total variation $d_{\mathrm{TV}}$ and KL divergence $d_{\mathrm{KL}}$ fluctuate along experiments, meaning that they can not truly indicate the coverage difference expectation. $\Delta\mathbb{E}$ can not capture the coverage difference expectation either. Lastly, due to the lack of robustness to score scales and outliers, standard, normalized, and truncated W-distance, denoted as $d_{\mathrm{W}}$, $d_{\mathrm{NW}}$, and $d_{\mathrm{TW}}$ respectively, can

222  not indicate the coverage difference expectation as well as $d_{\mathrm{NTW}}$. It also displays the average and
223  standard deviation of the Pearson coefficient of the proposed NTW and six baselines. NTW not only
224  has the highest average Pearson coefficient but also has the lowest standard deviation, which means
225  the correlation between NTW and the coverage difference expectation caused by concept shift is
226  very stable. In Figure 3 and Figure 4, we also visually show the correlation between the expected
227  coverage difference under concept shift and each metric. We refer to Appendix E for detailed analysis.
228  This observation suggests the potential of incorporating NTW in the training process, leading to the
229  development of the mRCP approach. By applying the NTW metric, mRCP aims to enhance coverage
230  robustness in test domains.

### 5.3  Experiments of mRCP

232  Since we prove NTW can assess expected coverage difference under concept shift effectively, mRCP
233  is designed to minimize it during training. In this case, validation sets are unnecessary, and we only
234  draw training, and test sets from $Q_{XY}^{(e)}$. Again, we draw one calibration set from $P_{XY}$. The model $f_\theta$
235  can also be an MLP or PDE based on different experiment setups. The loss function $l$ is the $\ell_1$ norm.
236  We implement mRCP according to Algorithm 1.

237  **Baselines:** Two methods of optimization with out-of-distribution data are selected as baselines. **DRO**
238  in Eq. (20) by [26] follows the minimax principle to reduce the highest $\mathcal{L}^{(e)}$ to obtain fair prediction
239  among test distributions. On the other hand, **V-REx** in Eq. (21), introduced by [18], focuses on
240  reducing the variance of $\mathcal{L}^{(e)}$ to obtain fairness. As we include importance weighting in mRCP, we
241  do not take it as a baseline, and the effectiveness of importance weighting is discussed in Section 6.

$$\mathcal{L}_{\mathrm{DRO}}(\theta) = \max_{e \in \mathcal{E}} \mathcal{L}^{(e)}. \tag{20}$$

242

$$\mathcal{L}_{\mathrm{V\text{-}REx}}(\theta) = \sum_{e \in \mathcal{E}} \mathcal{L}^{(e)} + \beta \operatorname{Var}(\mathcal{L}^{(e)} \mid e \in \mathcal{E}). \tag{21}$$

243  **Metric:** Denote $\mathbb{E}'_e[\mathbb{E}_\alpha[D^{(e)}]]$ the **expectation of coverage difference** over confidence levels
244  and test domains and $\mathbb{E}'_e[\mathcal{L}^{(e)}]$ the **expectation of prediction residual** over test domains. The
245  two expectations become **smaller** means the algorithm's performance is **better**. Both values are
246  normalized by the corresponding results from the same experiment setup trained by ERM. Changing
247  the weight $\beta$ in Eq. (19) will draw a Pareto front, thus we want the Pareto front closer to the origin.
248  Since V-REX is also controlled by a hyperparameter, we draw Pareto fronts for it as well.

249  **Result:** Figure 2 displays the Pareto fronts for mRCP, DRO, and V-REx, highlighting the trade-offs
250  between prediction residual and coverage difference expectation across different models and datasets.
251  Figure 2, (a) shows the results for the airfoil self-noise dataset when trained with a Multilayer
252  Perceptron (MLP) model. The mRCP method achieves a more favorable Pareto front compared to
253  V-REx, indicating a better balance between prediction residual and coverage difference expectation.
254  Additionally, mRCP attains a lower normalized coverage difference expectation than DRO at a
255  comparable level of the prediction residual. In Figure 2, (b), we observe the experiment results on
256  the epidemic spread prediction task using three epidemic datasets. With the same MLP architecture,
257  mRCP delivers superior Pareto fronts relative to the baselines. When employing the epidemic PDE,
258  the SIR model only has two trainable parameters, so their data points can not compose Pareto curves
259  due to the model's limited flexibility. Thus, we show the average of these points. Despite this
260  limitation, mRCP maintains its advantage over the baseline methods. Figure 2, (c) and (d) present
261  results from the traffic prediction task on three different traffic datasets. Here, the Pareto curves
262  for both the MLP and the reaction-diffusion (RD) PDE model are well-defined, because RD model
263  with six parameters, offers greater adaptability, allowing for clearer Pareto fronts. Overall, Figure 2
264  collectively indicates that mRCP consistently achieves lower coverage difference expectations without
265  compromising prediction residual as significantly as DRO and V-REx in different tasks and datasets.

## 6  Discussion

267  **mRCP can distinguish coverage differences under concept shift and covariate shift.** A notable
268  feature of the mRCP Pareto curves depicted in Figure 2 is their results when $\beta$ is small, which are not
269  at $\mathbb{E}'_e[\mathbb{E}_\alpha[D^{(e)}]] = 1$, unlike the Pareto curves of V-REx. This is because, during training, mRCP

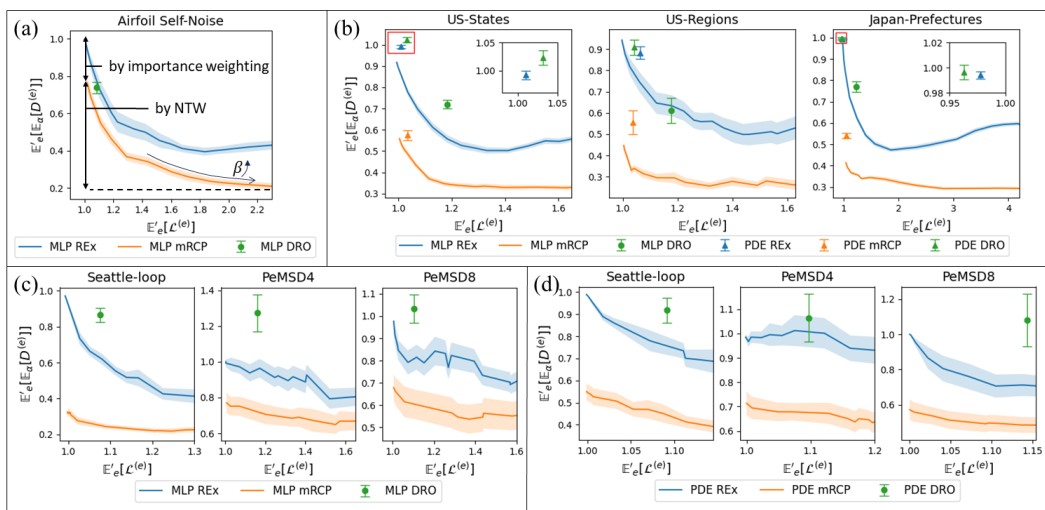

Figure 2: **Pareto fronts of Multi-domain Robust Conformal Prediction(mRCP), compared with DRO and V-REx:** Experimental results of (a) airfoil self-noise example, (b) epidemic spread prediction, and (c) (d) traffic speed prediction. mRCP always reaches a smaller coverage difference expectation than DRO and V-REx with less increase in prediction residual. Red boxes in (b) are zoomed-in areas. Shadow areas and error bars indicate the standard error.

has considered the coverage difference under covariate shift by applying importance weighting to calibration conformal score CDF. Consequently, as $\beta$ in Eq. (19) increases, the NTW term is only trained to mitigate the coverage difference under the concept shift, as shown in Figure 2,(a).

**DRO and V-REx are defeated because of improper selection of optimization metrics.** Examining Eq. (20) and Eq. (21), we can see both baselines aim to promote fairness by equalizing the expected losses across different domains. As the loss function is $\ell_1$ norm, which is identical to how conformal scores are calculated, the experiment results of $\Delta\mathbb{E}$ in the last row of Figure 3 show this metric is ineffective in capturing the coverage difference due to concept shift.

**Nonetheless, mRCP's limitations arise from the inherent challenges associated with penalty-based optimization algorithms.** Whether it is mRCP or V-REx, penalty-based optimization algorithms necessitate a model with a high capacity for fitting complex patterns. For instance, in Figure 2, (b), the Pareto curves are not discernible when predictions are derived from an epidemic PDE (SIR model) with only two adjustable parameters. In contrast, as shown in Figure 2, (d), the traffic PDE (RD model) demonstrates greater flexibility and adaptability with six tunable parameters, exhibiting distinct Pareto curves.

## 7 Conclusion

This study begins by decomposing the coverage difference caused by covariate and concept shifts. We then introduce the Normalized Truncated Wasserstein distance (NTW) as a metric for capturing coverage difference expectation under concept shift by comparing the test and weighted calibration conformal score CDFs. This metric can indicate the discrepancy position in calibration and test score distributions. Normalization and truncation make the metric score scales and outliers. Finally, we develop an end-to-end algorithm called Multi-domain Robust Conformal Prediction (mRCP) that incorporates NTW during training, allowing coverage to approach confidence in all test domains.

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

## A    Related work

Table 2: Related works and mRCP

| Task | Number of Test Domains | Test Domain Property | Work |
|------|------------------------|----------------------|------|
| Adaptive Conformal Prediction under Exchangeability | 1 | Identical to Calibration Domain | [22, 23] |
| Conformal Prediction under Covariate Shift | 1 | Covariate Shift | [29, 14] |
| Multi-Domain Conformal Prediction | Multiple | Feature-stratified | [7, 11] |
| | | Covariate Shift | [15] |
| | | Joint Distribution Shift in Certain $F$-divergence Range | [33, 6] |
| | | Joint Distribution Shift | mRCP |

## B    Error bound for the assumption of identical coverages

According to the computation of $q$ and $q^*$ in Eq. (1) and Eq. (6), respectively, we can define the coverages in unweighted and weighted calibration score distributions as

$$\hat{F}_P(q) = \inf\left\{\hat{F}_P(v_i)|\hat{F}_P(v_i) \geq \lceil(1-\alpha)(n+1)\rceil/n, v_i \in V_c\right\},$$

$$\hat{F}_{Q/P}(q^*) = \inf\left\{\hat{F}_{Q/P}(v_i)|\hat{F}_{Q/P}(v_i) \geq \lceil(1-\alpha)(n+1)\rceil/n, v_i \in V_c\right\}.$$

Denoting $q_+ = \inf\{v_i|v_i \in V_c, v_i > q\}$ and $q_+^* = \inf\{v_i|v_i \in V_c, v_i > q^*\}$, we can bound $\hat{F}_P(q)$ and $\hat{F}_{Q/P}(q^*)$ as

$$\hat{F}_P(q) \in \left[\lceil(1-\alpha)(n+1)\rceil/n, \hat{F}_P(q_+)\right), \quad \hat{F}_{Q/P}(q^*) \in \left[\lceil(1-\alpha)(n+1)\rceil/n, \hat{F}_{Q/P}(q_+^*)\right).$$

Therefore, the absolute difference between $\hat{F}^*(q^*)$ and $\hat{F}(q)$ is bounded by

$$|\hat{F}_{Q/P}(q^*) - \hat{F}_P(q)| < \max\left(\hat{F}_{Q/P}(q_+^*) - \lceil(1-\alpha)(n+1)\rceil/n, \hat{F}_P(q_+) - \lceil(1-\alpha)(n+1)\rceil/n\right).$$

Especially, when the calibration set size $n$ is large enough (like having thousands of samples), $\hat{F}_{Q/P}$ and $\hat{F}_P$ will be quite smooth, the upper above will be even negligible, allowing us to assume $\hat{F}_{Q/P}(q^*) = \hat{F}_P(q)$.

## C    Upper bound of coverage difference under concept shift

In this section, we prove that the W-distance between a test and weighted calibration conformal score population CDF can establish an upper bound for coverage difference under concept shift.

As $D$ quantifies the absolute difference between $\hat{F}_{Q/P}$ and $\hat{F}_Q$ at a calibration conformal score, it can be constrained by an upper bound given by the Kolmogorov distance [12] defined as follows.

**Definition 2** (Kolmogorov Distance)**.** *If $F_1$ and $F_2$ are two cumulative distribution functions (CDFs), the Kolmogorov distance, $d_K$, is defined as the maximum absolute difference between the CDFs.*

$$d_K(F_1, F_2) = \sup_{v \in \mathbb{R}} |F_1(v) - F_2(v)|.$$

As $\hat{F}_{Q/P}$ and $\hat{F}_Q$ are empirical (not population) CDFs of weighted calibration and test conformal scores, the bounding relationship can be reformulated as

$$d_K(\hat{F}_Q, \hat{F}_{Q/P}) = \sup_{v \in V_c \cup V_t} |\hat{F}_Q(v) - \hat{F}_{Q/P}(v)| \geq \sup_{v \in V_c} |\hat{F}_Q(v) - \hat{F}_{Q/P}(v)| = \sup_{v \in V_c} |D(v)|. \quad (22)$$

The upper bound $d_K(\hat{F}_Q, \hat{F}_{Q/P})$ depends on the two conformal score sets $V_c$ and $V_t$, indicating that the inclusion of samples in $S_c$ and $S_t$ is likely to introduce variability in $d_K(\hat{F}_Q, \hat{F}_{Q/P})$. Nevertheless, we aim for an upper bound that is not reliant on specific samples and relies on the calibration and test conformal score **population** CDFs, $F_P$ and $F_Q$.

Firstly, we convert the upper limit in Eq. (22) into terms of $F_P$ and $F_Q$. Denoting the joint probability density function (PDF) of features and score in the calibration and test domain as $\mathbf{p}_{XV}$ and $\mathbf{q}_{XV}$ respectively, the corresponding continuous CDFs of conformal scores are illustrated as

$$F_P(v) = \int_0^v \int_{\mathcal{X}} \mathbf{p}_{XV}(u,t)dudt, \ \ F_Q(v) = \int_0^v \int_{\mathcal{X}} \mathbf{q}_{XV}(u,t)dudt, \tag{23}$$

where $\mathcal{X}$ is the space of the feature variable $X$.

PDFs of features in calibration and test domains, denoted as $\mathbf{p}_X$ and $\mathbf{q}_X$ respectively, are defined as

$$\mathbf{p}_X = \int_{\mathbb{R}} \mathbf{p}_{XV}(u,t)dt, \ \ \mathbf{q}_X = \int_{\mathbb{R}} \mathbf{q}_{XV}(u,t)dt. \tag{24}$$

To address the coverage difference due to covariate shift, importance weighting from [29] is rewritten as $w = \frac{\mathbf{q}_X}{\mathbf{p}_X}$. Also, normalization is unnecessary, because $w$ here is a correction function to transform the marginal distribution of $\mathbf{p}_X$ into $\mathbf{q}_X$. The weighted version of $\mathbf{p}_{XV}$ is denoted as $\mathbf{p}'_{XV} = w\mathbf{p}_{XV} = \mathbf{q}_X\mathbf{p}_{V|X}$, which can be applied to derive the weighted continuous CDF of calibration conformal score by

$$F_{Q/P}(v) = \int_0^v \int_{\mathcal{X}} \mathbf{p}'_{XV}(u,t)dudt = \int_0^v \int_{\mathcal{X}} \mathbf{q}_X(u)\mathbf{p}_{V|X}(u,t)dudt. \tag{25}$$

The Kolmogorov distance between $F_{Q/P}$ and $F_Q$ is $d_K(F_Q, F_{Q/P}) = \sup_{v \in \mathbb{R}} |F_Q(v) - F_{Q/P}(v)|$.

**Theorem 1** (Triangular Inequality for Kolmogorov Distance). *If $F_1$, $F_2$, and $F_3$ are three cumulative distribution functions (CDFs), their Kolmogorov distances follow this inequality:*

$$d_K(F_1, F_3) \leq d_K(F_1, F_2) + d_K(F_2, F_3).$$

*Proof.* Consider any point $x \in \mathbb{R}$, then we have $|F_1(x) - F_3(x)| \leq |F_1(x) - F_2(x)| + |F_2(x) - F_3(x)|$. This inequality holds due to the triangle inequality for absolute values. Now, taking the supremum over all $x$, we have $\sup_{x \in \mathbb{R}} |F_1(x) - F_3(x)| \leq \sup_{x \in \mathbb{R}} (|F_1(x) - F_2(x)| + |F_2(x) - F_3(x)|)$. Note that the right-hand side is not necessarily equal to the sum of the suprema of the individual terms, because the points at which the suprema of $|F_1(x) - F_2(x)|$ and $|F_2(x) - F_3(x)|$ are attained may be different. However, we know that for any $x$, $|F_1(x) - F_2(x)|$ is at most $d_K(F_1, F_2)$ and $|F_2(x) - F_3(x)|$ is at most $d_K(F_2, F_3)$. Therefore, $\sup_{x \in \mathbb{R}} |F_1(x) - F_3(x)| \leq d_K(F_1, F_2) + d_K(F_2, F_3)$. Since the left-hand side is the definition of $d_K(F_1, F_3)$, we can demonstrate that $d_K(F_1, F_3) \leq d_K(F_1, F_2) + d_K(F_2, F_3)$. $\square$

As Kolmogorov distance satisfies the triangular inequality theorem, as shown and proved in Theorem 1, the triangular inequality relationship can be expanded to

$$d_K(\hat{F}_Q, \hat{F}_{Q/P}) \leq d_K(F_{Q/P}, \hat{F}_{Q/P}) + d_K(F_Q, F_{Q/P}) + d_K(\hat{F}_Q, F_Q). \tag{26}$$

Secondly, the Kolmogorov distance between an empirical CDF and its corresponding population CDF can be constrained by Dvoretzky–Kiefer–Wolfowitz (DKW) inequality [20], defined in Definition 3.

**Definition 3** (Dvoretzky–Kiefer–Wolfowitz (DKW) Inequality). *If $F$ is a population cumulative distribution function (CDF), and $\hat{F}$ is an empirical CDF with $n$ samples of a random variable $X$, then for any $\epsilon \geq \sqrt{\frac{1}{2n} \ln 2}$, the following inequality holds.*

$$\Pr(d_K(\hat{F}, F) > \epsilon) \leq e^{-2n\epsilon^2}.$$

Based on Definition 3, saying $|V_c| = n$ and $|V_t| = m$, we can apply DKW inequality to $d_K(\hat{F}_{Q/P}, F_{Q/P})$ and $d_K(\hat{F}_Q, F_Q)$ as follows, for $\epsilon \geq \sqrt{\frac{1}{2n} \ln 2}$ and $\rho \geq \sqrt{\frac{1}{2m} \ln 2}$.

$$\Pr(d_K(\hat{F}_{Q/P}, F_{Q/P}) \leq \epsilon) > e^{-2n\epsilon^2}, \ \ \Pr(d_K(\hat{F}_Q, F_Q) \leq \rho) > e^{-2m\rho^2}.$$

If the two events $d_K(\hat{F}_{Q/P}, F_{Q/P}) < \epsilon$ and $d_K(\hat{F}_Q, F_Q) < \rho$ are independent, the inequality in Eq. (26) can be expanded in Eq. (27), which holds with at least probability $e^{-2(n\epsilon^2 + m\rho^2)}$. By applying DKW inequality, we successfully quantify the variability of $d_K(\hat{F}_Q, \hat{F}_{Q/P})$ in Eq. (22) as a form of a probable event, and use the population conformal score CDFs to limit the worst-case of coverage difference under concept shift.

$$d_K(\hat{F}_Q, \hat{F}_{Q/P}) \leq d_K(F_Q, F_{Q/P}) + \rho + \epsilon. \tag{27}$$

Finally, having established in Eq. (13) that the W-distance can serve as an estimator for coverage difference expectation, we explore whether Eq. (27) may similarly be bounded by this metric. The W-distance of the two population conformal score CDFs are explicitly shown as

$$d_W(F_Q, F_{Q/P}) = \int_{\mathbb{R}} |F_Q(v) - F_{Q/P}(v)| dv = \int_{\mathbb{R}} \left| \int_0^v \int_{\mathbb{R}} \mathbf{q}_{XV}(u,t) du dt - \int_0^v \int_{\mathbb{R}} \mathbf{p}'_{XV}(u,t) du dt \right| dv$$

$$= \int_{\mathbb{R}} \left| \int_0^v \int_{\mathbb{R}} \mathbf{q}_{XV}(u,t) du dt - \int_0^v \int_{\mathbb{R}} \mathbf{q}_X(u) \mathbf{p}_{V|X}(u,t) du dt \right| dv \tag{28}$$

According to [24], if the weighted calibration conformal score probability density function (PDF) has Lebesgue density bounded by $\mathcal{C}$, which means $\mathbf{p}'_V$ does not exceed $\mathcal{C}$, then for any test conformal score PDF $\mathbf{q}_V$, $d_K(F_Q, F_{Q/P})$ can be bounded as

$$d_K(F_Q, F_{Q/P}) \leq \sqrt{2\mathcal{C} d_W(F_Q, F_{Q/P})} \tag{29}$$

Finally, we can derive the upper limit of coverage difference under concept shift, $\sup_{v \in V_c} |D(v)|$, in Eq. (30) at least probability $e^{-2(n\epsilon^2 + m\rho^2)}$.

$$\sup_{v \in V_c} |D(v)| \leq d_K(\hat{F}_Q, \hat{F}_{Q/P}) \leq \sqrt{2\mathcal{C} d_W(F_Q, F_{Q/P})} + \epsilon + \rho \tag{30}$$

This property is attractive in that the maximum difference in coverage due to concept shift can also be constrained in relation to the W-distance of population score CDFs, denoted as $d_W(F_Q, F_{Q/P})$. Despite the unobservability of $d_W(F_Q, F_{Q/P})$, we can still estimate it using its empirical form, $d_W(\hat{F}_Q, \hat{F}_{Q/P})$.

Even though coverage guarantee on an arbitrary joint shift is almost impossible, Eq. (28) demonstrates robust conformal prediction is attainable if we can train a function reducing the discrepancy between calibration and test conformal score distributions. To be specific, $d_W(F_Q, F_{Q/P})$ can be reduced to zero as far as $\mathbf{p}_{V|X} = \mathbf{q}_{V|X}$. In other words, if we regard $\mathbf{p}_{XV}$ and $\mathbf{q}_{XV}$ as push-forward probability distribution of $P_{XV}$ and $Q_{XV}$ by the trained model $f$, making the concept shift between $\mathbf{p}_{V|X}$ and $\mathbf{q}_{V|X}$ smaller will reduce coverage difference expectation on test domain.

# D  Datasets, models, and experiment setups

Extensive experiments are conducted under 3 tasks with 7 datasets. Some tasks involve both black-box and physics-informed models to demonstrate the generalizability of NTW and mRCP.

## D.1  Airfoil self-noise example

The airfoil dataset from the UCI Machine Learning Repository [5] consists of 1503 instances of 1-dimensional target $Y$ and 5-dimensional feature $X = (X_1, X_2, X_3, X_4, X_5)$. This dataset is manually separated and modified to create three different domains.

**Domain separation:**

Step 1. Covariate Shift by Data Separation. The original dataset is initially segmented into three primary subsets $A, B, C$ based on the 33% and 66% quantiles of the first dimension $X_1$. Subsequently, each of these subsets is further divided into three smaller portions at a 7:2:1 ratio, denoted like $A_{0.7}, A_{0.2}, A_{0.1}$ from $A$. Finally, we assemble three new datasets with covariate shift as $S^{(e_1)} = A_{0.7} \cup B_{0.2} \cup C_{0.1}$, $S^{(e_2)} = A_{0.2} \cup B_{0.1} \cup C_{0.7}$, $S^{(e_3)} = A_{0.2} \cup B_{0.1} \cup C_{0.2}$.

Step 2. Concept Shift by Target Modification. Differently distributed random noises are added to target values to cause concept shifts. For $y_i$ from $S^{(e_1)}$, $y_i+ = y_i/1000 * \tau$; for $y_i$ from $S^{(e_2)}$, $y_i+ = y_i/\tau$; for $y_i$ from $S^{(e_3)}$, $y_i+ = \tau$. $\tau$ follows a normal distribution $N(0, 10^2)$. Since we obtain three subsets in the end, $|\mathcal{E}| = 3$.

**Model selection:**

We utilize a straightforward multilayer perceptron (MLP) as a trainable model, with an architecture of (input dimension, 64, 64, 1) tailored for the regression task.

## D.2 Traffic speed prediction

The Seattle-loop [9], and PeMSD4, PeMSED8 datasets [16] contain sensor-observed traffic volume and speed data collected in Seattle, San Francisco, and San Bernardino. The snapshots from sensors are taken at 5-minute intervals. This task aims to predict the traffic speed of the local road segment in the next time step, using the traffic data from local and neighboring segments collected currently.

**Domain separation:**

Naturally, instances can be categorized into 24 subsets, $|\mathcal{E} = 24|$, based on the hour they are obtained. It is anticipated that there are joint shifts between the data distribution of every single hour (test domains) and the data distribution of the whole day (calibration domain), as traffic patterns vary over time, making it unnecessary to modify any data. We select the workday data from the three datasets.

**Model selection:**

(a) MLP with the same structure (input dimension, 64, 64, 1) is applied to the traffic prediction task.

(b) The Reaction-Diffusion (RD) model is selected as the physics-informed Partial differential equation (PDE) for traffic speed prediction. Reaction-diffusion mechanism, originally formulated for chemical systems to describe particle dynamics, has been adapted for traffic analysis by [3] to uncover traffic patterns on different road segments, offering an alternative to purely data-driven models like long-short-term memory. [28] further advanced this approach by integrating the RD model into graphical neural networks to capture traffic state interactions among adjacent road segments, with the reaction term accounting for influences against traffic flow and the diffusion term for influences along it. To be specific, for a given sensor $i$, with $N^d$ upstream and $N^r$ downstream neighboring sensors, the traffic states from these sensors impact sensor $i$ after $\delta t$ time through diffusion and reaction effects, respectively. We expand the original RD model in [28] to Eq. (31), where the traffic speed and volume at sensor $i$ at time $t$ is $u_i(t)$ and $q_i(t)$, respectively. The parameters $\rho_{(i,j)}$ and $\sigma_{(i,j)}$ represent the diffusion and reaction strengths between sensor $i$ and sensor $j$, while their superscripts indicate if they serve for speed or volume. Also, $d_i$ and $r_i$ are bias terms for the two components.

$$u_i(t + \delta t) - u_i(t) = \sum_{j \in N^d} (\rho_{(i,j)}^u(u_i(t) - u_j(t)) + \rho_{(i,j)}^q(q_i(t) - q_j(t)) + d_i$$
$$+ \tanh(\sum_{j \in N^r} \sigma_{(i,j)}^u(u_i(t) - u_j(t)) + \sigma_{(i,j)}^q(q_i(t) - q_j(t)) + r_i). \quad (31)$$

## D.3 Epidemic spread prediction

Three epidemic datasets, US-Regions, US-States, and Japan-Prefectures [10] include the number of patients infected by influenza-like illness (ILI) recorded by U.S. Department of Health and Human Services, Center for Disease Control and Prevention (CDC), and Japan Infectious Diseases Weekly Report. We aim to use the local population, the rise in the number of infected patients observed this week, and the cumulative total of infections as predictive features of the increase in infections for the upcoming week.

**Domain separation:** According to the Pandemic Intervals Framework (PIF) by CDC, samples are divided by four pandemic intervals, Initiation, Acceleration, Declaration, and Subsidence, so $|\mathcal{E}| = 4$. We establish the interval endpoints based on specific percentages of the total infected patient count, specifically at the 15%, 50%, and 85% thresholds.

**Model selection:**

(a) MLP with the same architecture is utilized for the epidemic spread forecasting task as well.

(b) PDE for this task is the SIR model that categorizes the population into three groups: those susceptible to the disease $S$, those infectious $I$, and those who have recovered and gained immunity $R$. It outlines the temporal changes in their populations, as described by [8]. The governing differential equations can be expressed as Eq. 32, where $N$, $\lambda$, and $\gamma$ represent the total population, infection rate, and recovery rate, respectively.

$$\begin{cases} \frac{dS(t)}{dt} = \frac{-\lambda S(t)I(t)}{N}, \\ \frac{dI(t)}{dt} = \frac{\lambda S(t)I(t)}{N} - \gamma I(t) = (\frac{\lambda S(t)}{N} - \gamma)I(t), \\ \frac{dR(t)}{dt} = \gamma I(t). \end{cases} \quad (32)$$

We make the assumption that the location is isolated, hence $N = S(t) + I(t) + R(t)$. Additionally, the population of recovered individuals is represented by $R(t) = \gamma \int_0^t I(t)dt$. Given this, if $t_o$ signifies the initial time of the current epidemic and $\delta t$ denotes the time step, which is a week in the three datasets, we can express the dynamic change of infectious individuals discretely as Eq. (33).

$$I(t + \delta t) - I(t) = \left( \frac{\lambda(N - I(t) - \gamma \sum_{t_o}^t I(t))}{N} - \gamma \right) I(t). \quad (33)$$

### D.4 Experiment setups for NTW and baseline metrics

As we only need to validate the positive correlation between NTW and coverage difference expectation, all models are trained by ERM. In the airfoil self-noise example, 100 trials are carried out. For the traffic task, 61 locations from the Seattle-loop, 59 locations from PeMSD4, and 33 locations from PeMSD8 are chosen, with 10 trials conducted at each location. For simplicity in the calculation, all selected locations have just one segment upstream and one segment downstream. For epidemic datasets, all locations from US-Regions, US-States, and Japan-Prefectures (49 locations in US-States, 10 locations in US-Regions, and 46 locations in Japan-Prefectures) are encompassed in the experiments, with 10 trials implemented on each location. The same experiment setups are operated on all baseline metrics and NTW. $\sigma$ values for MLP and PDE are 0.8 and 0.95, respectively. The ratio of training, calibration, validation, and testing data on airfoil self-noise datasets, three traffic datasets, and three epidemic datasets are 1:1:1:1, 3:2:2:3, and 1:2:1:1, respectively. Data separation was conducted randomly. Adam optimizer with a learning rate of 0.001 was applied for all experiments. On average, one trial requires one hours on a workstation with double NVIDIA RTX 3090 GPU.

### D.5 Experiment setups for mRCP, V-REx, and DRO

We define 1 trial as running a series of experiments of all predefined $\beta$ values once, except for DRO. For the airfoil self-noise example, 100 trials with random data preprocessing are conducted. For the traffic speed prediction task, we randomly select 10 locations from each of the three traffic datasets and operate one trial on all selected locations. In the epidemic spread prediction task, all locations of the three datasets are included and we operate one trial on each of them. All combinations of models (MLP and PDE) and algorithms (mRCP, DRO, V-REx) share the same experiment setups mentioned above. $\sigma$ values for MLP and PDE are 0.8 and 0.95, respectively. $\beta$ values for mRCP and V-REx in different experiment setups are shown in Table 3. Each Pareto curve consists of at least 10 $\beta$ values. For airfoil self-noise datasets and three traffic datasets, the original data is evenly and randomly split for training, calibration, and testing. For three epidemic datasets, we randomly split the original data for training, calibration, and testing with a ratio of 2:1:2. Adam optimizer with a learning rate of 0.001 was applied for all experiments. On average, one trial requires 12 hours on a workstation with double NVIDIA RTX 3090 GPU.

Table 3: $\beta$ values for mRCP and V-REx in experiment setups

| Dataset | Model | Algorithm | $\beta$ Values |
|---|---|---|---|
| Airfoil Self-Noise | MLP | mRCP | 0.1, 0.2, 0.5, 1, 2, 5, 10, 15, 20, 30, 50, 80, 100. |
| | | V-REx | 0.1, 1, 2, 2.5, 3, 3.5, 4, 4.5, 5, 6, 7, 8, 9, 10, 15, 20. |
| Japan-Prefectures | MLP | mRCP | 0.1, 0.2, 0.4, 0.8, 1, 2, 5, 10, 20, 40, 100, 200, 500. |
| | | V-REx | 0.1, 1, 1.5, 2, 3, 4, 5, 7.5, 10, 20, 40, 100, 200, 500. |
| | PDE | mRCP | 0.1, 0.2, 0.4, 0.6, 0.8, 1, 2, 5, 7, 10. |
| | | V-REx | 0.1, 0.2, 0.4, 0.6, 0.8, 1, 2, 5, 7, 10. |
| US-Regions | MLP | mRCP | 0.1, 0.2, 0.4, 0.8, 1, 2, 5, 10, 20, 40, 100, 200, 500. |
| | | V-REx | 0.1, 1, 2, 3, 4, 5, 6, 7, 8, 10, 15, 20, 30, 40, 100. |
| | PDE | mRCP | 0.1, 0.2, 0.4, 0.6, 0.8, 1, 2, 5, 7, 10. |
| | | V-REx | 0.1, 0.2, 0.4, 0.6, 0.8, 1, 2, 5, 7, 10. |
| US-States | MLP | mRCP | 0.1, 0.2, 0.4, 0.8, 1, 2, 5, 10, 20, 40, 100, 200, 500. |
| | | V-REx | 0.1, 1, 1.2, 1.7, 2, 2.5, 3, 3.5, 4, 5, 7, 10, 15. |
| | PDE | mRCP | 0.1, 0.2, 0.4, 0.6, 0.8, 1, 2, 5, 7, 10. |
| | | V-REx | 0.1, 0.2, 0.4, 0.6, 0.8, 1, 2, 5, 7, 10. |
| Seattle-loop | MLP | mRCP | 0.1, 1, 2, 3, 4, 5, 10, 25, 100, 200, 400, 700, 1000. |
| | | V-REx | 0.1, 1, 1.5, 2, 2.5, 3, 4, 5, 10, 50. |
| | PDE | mRCP | 0.1, 1, 5, 10, 20, 40, 80, 160, 320, 640. |
| | | V-REx | 0.1, 0.2, 0.5, 0.8, 1, 1.5, 2, 3, 4, 5. |
| PeMSD4 | MLP | mRCP | 0.1, 1, 2, 5, 10, 50, 100, 150, 200, 300, 400, 500. |
| | | V-REx | 0.1, 1, 2, 3, 4, 5, 7.5, 10, 13, 16, 19, 22, 25. |
| | PDE | mRCP | 0.1, 1, 5, 10, 50, 100, 200, 500, 1000, 5000, 10000. |
| | | V-REx | 0.1, 1, 2, 3, 4, 5, 7, 8, 10, 12, 15. |
| PeMSD8 | MLP | mRCP | 0.1, 1, 2, 5, 10, 50, 100, 150, 250, 300, 400, 500. |
| | | V-REx | 0.1, 1, 2, 3, 4, 5, 7.5, 10, 20, 30, 40, 75, 80, 150. |
| | PDE | mRCP | 0.1, 1, 5, 10, 50, 100, 200, 500, 1000, 2000. |
| | | V-REx | 0.1, 1, 2, 3, 5, 7, 10, 15, 20, 30. |

# E   Additional experiment results

## E.1   Pearson coefficient definition

Here we provide a detailed definition of the Pearson coefficient as follows.

**Definition 4** (Pearson coefficient). *The Pearson correlation coefficient, denoted as $r$, is calculated as the covariance of the two variables divided by the product of their standard deviations, as follows.*

$$r = \frac{\sum (x_i - \overline{x})(y_i - \overline{y})}{\sqrt{\sum (x_i - \overline{x})^2 \sum (y_i - \overline{y})^2}}. \tag{34}$$

*where $x_i$ and $y_i$ are the individual sample points of random variables $X$ and $Y$ indexed with $i$ and $\overline{x}$ and $\overline{y}$ are the means of their samples, respectively.*

The Pearson correlation coefficient measures the linear correlation between two variables. It gives a value between -1 and 1 inclusive, where 1 indicates a perfect positive linear relationship, -1 indicates a perfect negative linear relationship, and 0 indicates no linear correlation.

## E.2   Correlation visualization

Figure 3 shows the experimental results of the correlation between NTW and coverage difference expectation, compared with three baselines: total variation, KL divergence, and expectation difference. It is organized into a matrix of subplots, with each column corresponding to a specific dataset and each row depicting the performance of a metric. Within these subplots, individual points represent the conjunction of a metric's value with the associated coverage difference expectation for a given test domain. A positive trend between NTW and the coverage difference expectation is shown in the

top row, showcasing NTW's strong correlation. In contrast, the other metrics exhibit inconsistent correlations across the varied datasets and models, as seen in the lower three rows of subplots. Figure 4 also illustrates the expected coverage difference's correlation to NTW, standard W-distance, normalized W-distance, and truncated W-distance, proving that normalization and truncation are equally important for robust correlations.

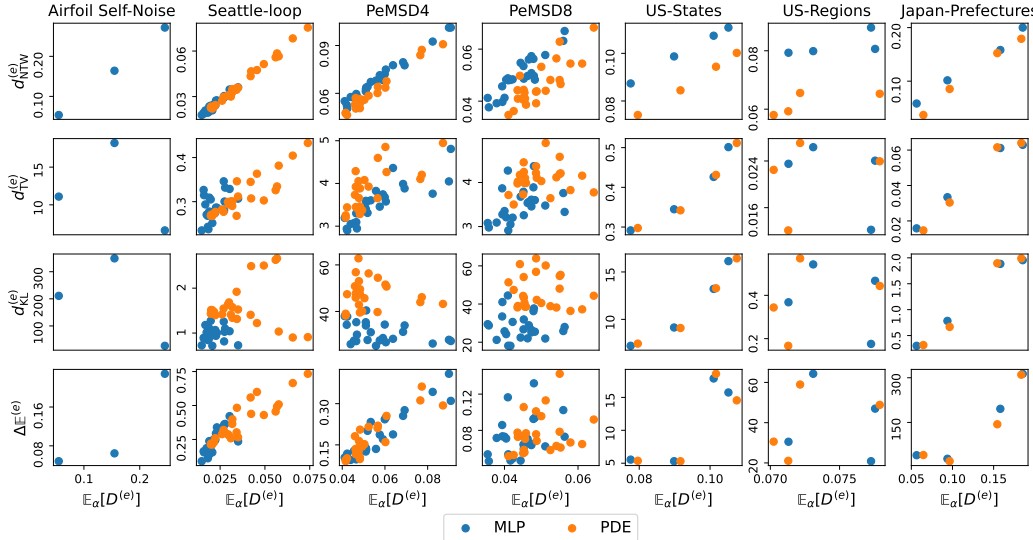

Figure 3: **Experimental results of the correlation between Normalized Truncated Wasserstein distance (NTW) and coverage difference expectation, compared with total variation, KL divergence, and expectation difference**. Each point represents a pair of metric value and coverage difference expectation for a test domain. The first row of the subplots demonstrates NTW indicates the expectation across different datasets and models, whereas other baseline metrics, represented in the other three rows, can not consistently capture it.

575

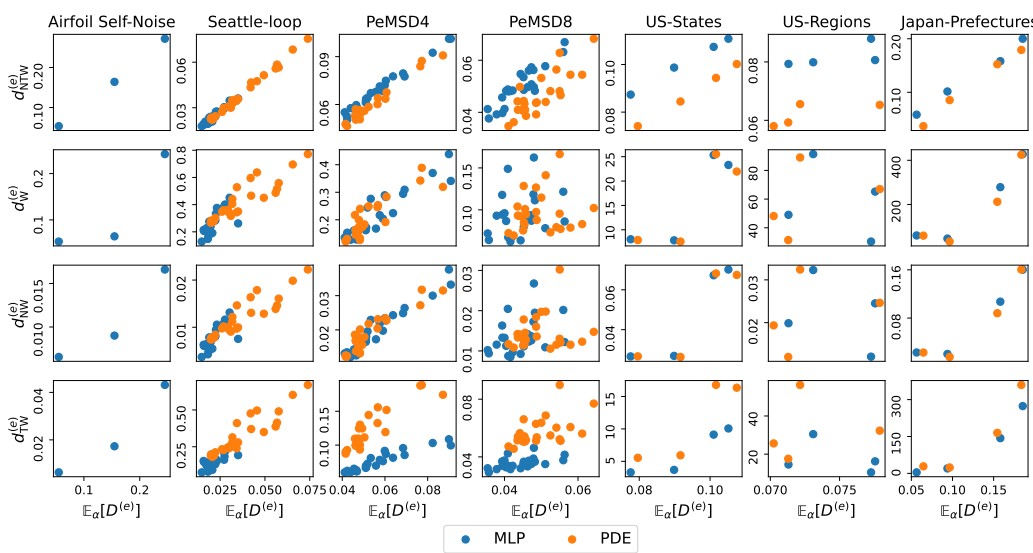

Figure 4: **Experimental results of the correlation between Normalized Truncated Wasserstein distance (NTW) and coverage difference expectation of concept shift, compared with standard, normalized, and truncated Wasserstein distance**. Each point represents a pair of metric value and coverage difference expectation of a test domain. By comparing the first row with the rest three rows, we validate the necessity of applying normalization and truncation together.

