# OpenReview forum: "Robust Conformal Prediction under Joint Distribution Shift"
_NeurIPS.cc/2024/Conference — Submitted to NeurIPS 2024_

### Official Review · Reviewer_m45f · 2024-07-06

**Soundness:** 2
**Presentation:** 3
**Contribution:** 2
**Rating:** 5
**Confidence:** 3

**Summary:**

This paper adresses the issue of conformal prediction under distribution shift with multiple test domains. The goal is to reduce the deviation in coverage caused by different potential distribution shifts across these domains. The paper firstly proposes a way of disentangling a joint distribution shift (shift effects both in covariate and label distributions, which they term "concept") by employing weighted conformal prediction (Tibshirani et al, 2019) to address covariate shift, and then quantify the remaining shift in terms of a truncated normalized Wasserstein distance (D-NTW) between the original and weighted conformal score distributions (empirical CDFs). This D-NTW is then used as a regularizer term in a training algorithm to explicitly ensure that coverage deviations are minimized across test domains. Experiments include assessing the correlation between D-NTW and the actual expected coverage difference, which is shown to be high (vs. other distributional distance metrics), and comparing to two multi-test-domain optimization methods on a variety of datasets to show that the coverage difference is lower while not compromising on prediction accuracy.

**Strengths:**

- Addressing the issue of distribution shift in conformal prediction is a relevant problem, particularly for the less explored label shift setting
- Attempts are made to disentangle covariate and label (concept) shifts, which can provide insights into the model's adaptation abilities
- The suggested D-NTW distance metric is well motivated and benchmarked against multiple sensible alternative distance metrics
- An interesting range of datasets from different domains is considered
- The paper is clearly written and good to follow, including Fig I visualizing the procedure (albeit it is somewhat hard to read, especially part (c))

**Weaknesses:**

My main concerns are w.r.t. practicality and evaluation. A fundamental requirement of the proposed algorithm and D-NTW is the availability of labelled samples from every test domain $Q_{XY}^{(e)}$ in order to obtain the conformal test score distributions $F_{Q^{(e)}}$, from which then both the likelihood ratios for covariate shifts and D-NTW can be explicitly computed. Beyond existing concerns about the practicality of estimating a likelihood $\textit{ratio}$, we now require actually explicitly estimating every test domain distribution, which is prone to much error. Regardless, if we can now explicitly obtain $F_{Q^{(e)}}$ for every test domain, I am wondering why the direct solution is not to just compute a conformal quantile $q^{(e)}$ on the basis of this for every test domain and thus optimally adapt to the existing joint shifts per domain. Perhaps the loss of the disentanglement of shift contributions is the motivation? In general, I find this requirement to require knowing or estimating the test domain distributions on the basis of labelled test domain data and thus the use of extensive density estimation quite limiting, especially if we consider high-dimensional settings. Perhaps this is why the considered experiments are for 1-D regression data only. I was also wondering if the authors were able to make any connections between their proposals and obtainable coverage guarantees on the test domains. They propose a bound on the distance of D-NTW from the expected coverage difference, but perhaps more explicit connections to conformal coverage guarantees of the form in Eq. 3, e.g. by leveraging guarantees from (Tibshirani et al, 2019) and their linear decomposition of shift effects are worth investigating.

In regards to evaluation, I was missing a closer connection to existing conformal methods under shift, and the actual goals of these methods in terms of coverage guarantees on test data. In their comparison to test-domain optimization algorithms I was not surprised that their algorithm performs better on expected coverage difference, since it explicitly $\textit{optimizes}$ for this goal, while the baselines target e.g. variance minimization. It would be more interesting to compare to conformal algorithms for shift such as the mentioned [2,3], showing e.g. that those are not able to fully capture the joint shift or are overly conservative, thus compromising on the metric. For example, I was surprised that [1] was not mentioned, since it explicitly targets label shifts. Similarly, while it is nice that the relative coverage difference is minimized or correlates well with D-NTW, this does not tell explicitly how my conformal methods will now perform on the test domains. It would be nice to also obtain an explicit assessment of the coverage $(1-\alpha)$ on test domains, and the obtained prediction set sizes. Even if target coverage is not satisfied, it would already be a contribution to show that the proposed algorithm achieves better robustness by being closer to target coverage, or smaller set sizes at the same level.

Minor: multiple typos e.g. L75, Fig I caption, L89, Eq. 2

References
- [1] Podkopaev, Aleksandr, and Aaditya Ramdas. "Distribution-free uncertainty quantification for classification under label shift." Uncertainty in artificial intelligence. PMLR, 2021.
- [2] Cauchois, Maxime, et al. "Robust validation: Confident predictions even when distributions shift." Journal of the American Statistical Association (2024): 1-66.
- [3] Zou, Xin, and Weiwei Liu. "Coverage-Guaranteed Prediction Sets for Out-of-Distribution Data." Proceedings of the AAAI Conference on Artificial Intelligence. Vol. 38. No. 15. 2024.

**Questions:**

- Could you comment on some of the concerns raised above, e.g. on the practicality of requiring multiple density estimations, simply computing optimal quantiles directly on each test domain, or evaluating the methods w.r.t. test coverage and conformal set size.
- What methods are used to estimate the test domain distributions $Q_{XY}^{(e)}$?
- Have you considered higher-dimensional experimental settings and/or using models that are more complex than an MLP? If not, could you explain if this is related to the limitations of your method in terms of higher dimensions?
- In the experiments, all the generated test domain shifts are created manually. Have you considered datasets with unknown test shifts, e.g. the Yearbook dataset?
- In sec 5.2. the correlation is assessed via Pearson coefficient. Could you provide a reasoning on why you believe a linear relationship exists, and/or if you have considered more robust measures such as rank correlation? Given that the estimated score distributions are empirical CDFs, this seems perhaps more intuitive.
- Could you comment on the observed discrepancies in some of the results in Table 1? For example, For US-States and MLP, all distance metrics except D-NTW show negative correlation.
- Could you comment on the limitations of the made assumption in Eq. 17 that the calibration domain $P_{XY}$ is considered a linear mixture of the "unknown" test domain distributions? This seems like a restriction that is not explicitly mentioned.

**Limitations:**

The requirements of the algorithm and density estimations are mentioned, but the limitations of the approach are not explicitly discussed in terms of imposed assumptions on the problem setting. A more through discussion of the practical limitations would be helpful (some of which are inherited e.g. from (Tibshirani et al, 2019) simply by their use of likelihood ratio weights).  A small subsection in sec 6 mentions difficulties of optimization algorithms.

---

> ### Author Rebuttal · Authors · 2024-08-07
>
> Thank you for your questions.
> >My main concerns are w.r.t. practicality and evaluation. A fundamental requirement of the proposed algorithm and D-NTW is the availability of labelled samples from every test domain $Q _{XY}^{(e)}$ in order to obtain the conformal test score distributions, from which then both the likelihood ratios for covariate shifts and D-NTW can be explicitly computed. Beyond existing concerns about the practicality of estimating a likelihood ratio, we now require actually explicitly estimating every test domain distribution, which is prone to much error.
>
> We introduce how we applied kernel density estimation (KDE) to estimate the likelihood ratio in the author rebuttal. Indeed, the process of KDE is prone to error with a limited sample size from each test domain, especially for high-dimension regression. The experiment of the Airfoil dataset has the smallest sample size, 160, of one test domain, and the highest regression dimension, 5. Figure 2 (a) shows the experiment result of that dataset, and the performance is acceptable. Nevertheless, it is still challenging for KDE with higher dimensions and fewer samples.
>
> > Regardless, if we can now explicitly obtain $\hat{F}_Q^{(e)}$ for every test domain, I am wondering why the direct solution is not to just compute a conformal quantile q(e) on the basis of this for every test domain and thus optimally adapt to the existing joint shifts per domain.  Perhaps the loss of the disentanglement of shift contributions is the motivation?
>
> The reason why we do not optimize based on joint distribution shift but on concept shift is because covariate shift can be addressed by importance weighting, even if estimation error can be introduced by KDE. Intuitively, since a model has limited fitting ability, we want it to focus on the issue (concept shift) that cannot be addressed by importance weighting to avoid wasting its representativeness.
>
> We do not compute a quantile for each $\hat{F}_Q^{(e)}$ and adapt to them, because a quantile value is $(1-\alpha)$-specific, but we want the trained model to be comprehensive, so test coverage approaches $1-\alpha$ no matter how $\alpha$ changes.
>
> >In general, I find this requirement to require knowing or estimating the test domain distributions on the basis of labelled test domain data and thus the use of extensive density estimation quite limiting, especially if we consider high-dimensional settings.
>
> For your concerns about high-dimensional settings, higher dimensions will make the KDE less reliable.  However, we do not only consider 1-d regression tasks. Line 60 may cause some confusion. In fact, $\mathcal{X}\subset\mathbb{R}^d$. For the airfoil dataset, d=5 in line 463. For three traffic datasets, d=4. For epidemic datasets, d=2.
>
> > It would be more interesting to compare to conformal algorithms for shift such as the mentioned [2,3], showing e.g. that those are not able to fully capture the joint shift or are overly conservative, thus compromising on the metric.
>
> To illustrate the benefits of mRCP, we provide experimental results about coverage and prediction intervals in comparison with the WC method [R1] in the author rebuttal.
>
> >Have you considered higher-dimensional experimental settings and/or using models that are more complex than an MLP?
>
> High-dimension regression tasks are included in experiments. As we mentioned, for the airfoil dataset, d=5. For three traffic datasets, d=4. For epidemic datasets, d=2. Applying mRCP to models with higher fitting ability may obtain better performance as they can have a better trade-off between residual loss and distribution discrepancy loss.
>
> >In the experiments, all the generated test domain shifts are created manually. Have you considered datasets with unknown test shifts?
>
> Except for the airfoil dataset, all three traffic datasets and three epidemic datasets have natural joint distribution shifts without any manual modification. Their joint shifts are unknown, and we can only estimate their covariate shift by KDE as mentioned above.
>
> >In sec 5.2. the correlation is assessed via the Pearson coefficient. Could you provide a reasoning on why you believe a linear relationship exists, and/or if you have considered more robust measures such as rank correlation? Given that the estimated score distributions are empirical CDFs, this seems perhaps more intuitive.
>
> We actually only expect a monotonic positive correlation because $\mathbb{E} _{\alpha}[D^{(e)}]$
> is the average of discrete $D^{(e)}$ values for $\alpha = 0.1,0.2,...,0.9$ and $d _{NTW}^{(e)}$ is the distributional distance. However, experimental results also show a strong positive linear correlation. That is because both $d _{NTW}^{(e)}$ and $\mathbb{E} _{\alpha}[D^{(e)}]$ approximate Eq. (10) well, so as Eq. (10) increases, $d _{NTW}^{(e)}$ and $\mathbb{E} _{\alpha}[D^{(e)}]$ increases proportionally.
>
> We present the result of **Spearman's rank correlation coefficient** in the author rebuttal.
>
> > For US-States and MLP, all distance metrics except D-NTW show negative correlation.
>
> Please take a look at Figure 3. US-States only have four test domains; thus, the coefficient can be negative if the four points do not show a positive correlation.
>
> >Could you comment on the limitations of the made assumption in Eq. 17 that the calibration domain $P_{XY}$ is considered a linear mixture of the "unknown" test domain distributions? This seems like a restriction that is not explicitly mentioned.
>
> mRCP is also applicable when $P_{XY}$ is not a linear mixture of the test domains. The role of $P_{XY}$ is to provide a ‘target’ conformal score distribution for all test score distributions so that they can approach. It is worth in study the performance of mRCP if $P_{XY}$ is a more complex combination of $Q_{XY}^{(e)}$
>
> **Reference**
>
> [R1] Cauchois M, Gupta S, Ali A, et al. Robust validation: Confident predictions even when distributions shift[J]. Journal of the American Statistical Association, 2024: 1-66.

---

> > ### Comment · Reviewer_m45f · 2024-08-11
> >
> > Thank you for your comments and clarifications.
> >
> > I believe conformal comparisons such as the additional experiments against Cauchois et al. are a stronger approach to validating the performance of your method rather than optimization baselines, and should be actively pursued. Similarly, it is good to see that the rank correlation relationships also hold, showing promise for some of your design choices such as the D-NTW distance.
> >
> > That being said, after having read some of the other rebuttals and acknowledging raised concerns with the theoretical positioning and motivation of the work (e.g., its empirical vs. theoretical validity), I prefer to keep my current score at this moment.

---

### Official Review · Reviewer_6oEU · 2024-07-07

**Soundness:** 3
**Presentation:** 3
**Contribution:** 2
**Rating:** 4
**Confidence:** 3

**Summary:**

This paper studies the coverage difference caused by covariate and concept shifts. Authors introduce the Normalized Truncated Wasserstein distance (NTW) as a metric for capturing coverage difference expectation under concept shift by comparing the test and weighted calibration conformal score CDFs. They also develop an end-to-end algorithm called Multi-domain Robust Conformal Prediction (mRCP) to incorporate NTW during training, allowing coverage to approach confidence in all test domains.

**Strengths:**

1. introduced NTW as a metric to capture the coverage gap

2. high correlation between NTW and coverage difference expectation; mRCP can balance residual size and coverage gap

**Weaknesses:**

1. section 3.1 and 3.2 introduce some important definitions:  while authors provide some explanation, the theoretical understanding of them are very limited

2. simulation:  authors mention the mRCP can achieve a balance between coverage gap and size of residual, further simulations need to be carried out ( I would be interested to see a plot including the avg. coverage vs avg. residual size

**Questions:**

3.1 the coverage difference works with the empirical distribution function, would it be possible to study the coverage gap in the form of P(...) - P(...)

3.1 in the paper, a likelihood ratio dQ/dP is assumed to be known, how would you estimate it in practice? The definition of $q^{*}$ in the paper seems to be inconsistent with the conformal prediction literature. The factor (n+1)/n, I believe is typically added in the i.i.d case while  for covariate shift, the form is not like this if you take a look at the paper "conformal prediction under covariate shift".


4  In practice, how would you select the tuning parameter beta? Better theoretical guarantees for section 4 need to be developed.

**Limitations:**

Authors mentioned the limitation in the discussion.

---

> ### Author Rebuttal · Authors · 2024-08-07
>
> Thank you for your questions.
> >section 3.1 and 3.2 introduce some important definitions: while authors provide some explanation, the theoretical understanding of them are very limited.
>
> Section 3.1 and 3.2 introduce the process of quantifying the empirical coverage gap caused by concept shift, and connecting it to a distributional discrepancy metric.
>
> **The theoretical analysis of bounding the coverage gap by population distributions with a probability related to the number of calibration and test samples is discussed in Appendix C**.
>
> >simulation: authors mention the mRCP can achieve a balance between coverage gap and size of residual, further simulations need to be carried out ( I would be interested to see a plot including the avg. coverage vs avg. residual size
>
> To further illustrate the **coverages obtained by mRCP**, we provide experimental results about coverage and prediction intervals in comparison with the WC method [R1] in the author rebuttal.
>
> >  the coverage difference works with the empirical distribution function, would it be possible to study the coverage gap in the form of P(...) - P(...)
>
> We developed the upper bound of the empirical coverage gap in **population form** in Appendix C.
>
> >in the paper, a likelihood ratio dQ/dP is assumed to be known, how would you estimate it in practice?
>
> **The likelihood ratio is not assumed known** and it is estimated by kernel density estimation. Please read the author rebuttal for details.
>
> >The factor (n+1)/n, I believe is typically added in the i.i.d case while for covariate shift, the form is not like this if you take a look at the paper "conformal prediction under covariate shift".
>
> Using the factor $(n+1)/n$, the quantile under exchangeability assumption is given by
>
> $$
> q=\text{Quantile}\left(\frac{\lceil(1-\alpha)(n+1)\rceil}{n},\frac{1}{n}\sum_{v_i\in V_c}\delta _{v_i}\right).
> $$
>
> In Lemma 1 of [R2], the quantile is calculated by
>
> $$
> q=\text{Quantile}\left(1-\alpha,\frac{1}{n+1}\sum_{v_i\in V_c}\delta _{v_i}+\delta _{\infty}\right).
> $$
>
> Both forms produce the same quantile value under the i.i.d. case.
>
> [R2] calculates the importance-weighted quantile based on the latter one, whereas we present Eq. (6) based on the former one. However, there is a minor difference when applying these two forms to importance weighting,  regarding how $p _i^w$ and $p _{n+1}^w$ are calculated in Eq. (7) of [R2].
>
>  In Eq. (7) of [R2], the $p _i^w$ and $p _{n+1}^w$ are functions of the test input $x$, as $\delta _{\infty}$ is included its Lemma 1. However, in our work, the weights of conformal scores are not functions of $x$.
>
> In Section 4.5 of [R3], another form is provided to get rid of the factor $(n+1)/n$ and $\delta _{\infty}$. We think that is a good example to represent the importance-weighted quantile and would like to take it in the later version.
>
> >In practice, how would you select the tuning parameter beta?
>
> $\beta$ is the hyperparameter for our proposed method in Eq (19). Therefore, we need to try different $\beta$ values to draw a Pareto front of the prediction residuals (horizontal axis) and coverage gap (vertical axis) in Figure 2. For each $\beta$ value, we train a model and obtain an optimal Pareto solution of (residual, coverage gap) and finally, we get curves in Figure 2. The baseline V-REx is also tuned by a hyperparameter, so we tried different $\beta$ for V-REx as well in Figure 2. We show our selected $\beta$ values in Table 3. The basic goal of selecting $\beta$ values is to make the Pareto solutions more uniformly distributed and dense enough to obtain reliable Pareto curves.
>
> >Better theoretical guarantees for section 4 need to be developed.
>
> When a theoretical coverage guarantee under distribution shift is obtained, it is very hard to keep prediction sets small at the same time. For instance, the worst-case (WC) method [R1] holds the guarantee under distribution shift at the cost of inefficient large prediction sets.
>
> The purpose of Section 4 is to get **a good trade-off between the coverage and prediction interval size**.  This good trade-off can be obtained because of two reasons. First, we do not follow conservative methods as mentioned above. Secondly, prediction interval size is highly related to the magnitude of conformal scores which is absolute residual for regression tasks, and Eq. (19) actually balances the residual loss and the NTW loss, which represents the gap to $1-\alpha$.
>
> In Figure 5 of the author rebuttal attachment, even if mRCP coverage is not at least $1-\alpha$, it approaches $1-\alpha$ very close.
>
> **Reference**
>
> [R1] Cauchois M, Gupta S, Ali A, et al. Robust validation: Confident predictions even when distributions shift[J]. Journal of the American Statistical Association, 2024: 1-66.
>
> [R2] Tibshirani R J, Foygel Barber R, Candes E, et al. Conformal prediction under covariate shift[J]. Advances in neural information processing systems, 2019, 32.
>
> [R3] Angelopoulos A N, Bates S. Conformal prediction: A gentle introduction[J]. Foundations and Trends® in Machine Learning, 2023, 16(4): 494-591.

---

### Official Review · Reviewer_qkU5 · 2024-07-18

**Soundness:** 1
**Presentation:** 2
**Contribution:** 2
**Rating:** 3
**Confidence:** 4

**Summary:**

The paper "Robust Conformal Prediction under Joint Distribution Shift" investigates the problem of predictive inference under the setting where we have both covariate shift and concept shift. The authors propose a conformal prediction-based procedure that accounts for such distribution shifts and illustrate the performance through experiments.

**Strengths:**

This work provides extensive and thorough experimental results.

**Weaknesses:**

The paper doesn't read very well as some notations appear without definitions and relevant assumptions are not written clearly. For example:

- It is assumed that the likelihood ratio $dQ/dP$ is known, but this was not clearly stated in the problem setting.
- Some assumptions are not written in advance but are rather introduced when they are needed.

It would be better if the authors could provide sufficient intuition and motivation for their methodology.

**Questions:**

1. First of all, as a theoretical statistician who is not very familiar with the standards in the machine learning community, I wonder what the motivation of this work is. Specifically, the main advantage of the conformal prediction framework is that it theoretically provides an exact finite sample guarantee without distributional assumptions. In this work, the proposed methods have no theoretical guarantee, and the focus is more on empirical illustrations. While I understand that machine learning emphasizes practical applications and performance on real data, why should one apply conformal prediction here if we cannot exploit the advantage of the conformal inference framework? In other words, If we are not aiming for a distribution-free guarantee (or any other theoretical guarantee), aren't there methods that work better for specific datasets?

2. Evaluating the difference under distribution shift using (5) doesn't seem very reasonable to me. The prediction set is constructed using the calibration set, and the authors estimate the 'CDF of the score under no distribution shift' by $\hat{F}_P$, which is also a function of the calibration set (though estimating the 'CDF under shift' with $\hat{F}_Q$ sounds reasonable).

3. In fact, $\hat{F}_P(q)$

  seems to be just $1-\alpha$ (more precisely, $\lceil(n+1)(1-\alpha)\rceil / n$)? According to the definition, $q$ is the quantile of the distribution $\frac{1}{n}\sum_{i=1}^n \delta_{v_i}$, while $\hat{F}_P$

is the CDF of the same distribution. So it seems to me that $\hat{F}_P(q) = \lceil(n+1)(1-\alpha)\rceil / n$ holds exactly.

 Then it is quite strange why they replace this known value with $\hat{F}_{Q/P}(q^*)$, generating an additional error.

4. Additionally, I'm a bit confused about equation (13), where again the exact equality seems to hold. The two empirical CDFs $\hat{F}_Q$

 and $\hat{F}_{Q/P}$ are step functions where the jumps occur at the same values $v_1, \cdots, v_n$, so it seems to me that exact equality holds. In that sense, the remaining discussion in section 3.2 sounds a bit odd. For example, the empirical CDFs have values exactly 1 for $v \geq \max v_i$. What does the 'long tail' mean in this case?

5. Could the authors clarify the motivation behind the need for 'multi-domain' inference? In the beginning of Section 4, it just says, "The domain $P_{XY}$ can be decomposed into $M$ multiple domains," so it's a bit unclear what settings are under consideration. For example, are they considering a setting where there is a natural partition of the domain, and we aim for a good quality of inference conditional on each element of the partition? Or are they considering a setting where we artificially construct a partition (if so, what is the reason for that?), etc.? This might be provided in previous literature, but it would be better if relevant discussions are also provided in this work.

6. The notation in equation (17) is hard to understand. The notation $P_{XY}$ was originally used to denote the distribution of $(X,Y)$, but the authors use it to denote the domain (in the line above equation (17)), and in equation (17) it is written as if the distribution is a mixture distribution of $M$ distributions $(Q_{XY}^{(e)})$ with mixture probability $(\frac{1}{M}, \cdots, \frac{1}{M})$, rather than the domain being partitioned into $M$ bins. Does $Q_{XY}^{(e)}$ denote the conditional distribution of $(X,Y)$ under $P_{XY}$, given $(X,Y) \in e$? If that's the case, how do we know that the probability $P((X,Y) \in e)$ is $\frac{1}{M}$ for every $e$ in the partition?

7. I wish I could come up with more constructive questions and suggestions, but it was a bit hard for me to do so unless the above points (which are related to the technical foundation of the main results) are resolved. I understand that some of the points above might be due to my misunderstanding or confusion, and I'd be happy to have a further discussion or update the scores once I receive a reply from the authors.

* Minor comments:

In equation (10), the notation `E' doesn't seem appropriate as it's a sample mean rather than the population mean.

I believe It is more standard to denote $v_i$ as 'nonconformity score' rather than 'conformal score.'

* Typos:

above equation (3): 'coverage guarantee'

Equation (4): $v$ -> $v_i$

Equation (11): $dx$ -> $dv$

**Limitations:**

I don't think this work has negative societal impacts.

---

> ### Author Rebuttal · Authors · 2024-08-07
>
> Thank you for your questions.
> >It is assumed that the likelihood ratio dQ/dP is known...
>
> **The likelihood ratio is not assumed known and it is estimated by kernel density estimation (KDE).** Please read the author rebuttal for details.
>
> > In this work, the proposed methods have no theoretical guarantee, and the focus is more on empirical illustrations. ... Why should one apply conformal prediction here if we cannot exploit the advantage of the conformal inference framework?
>
> Standard CP can provide coverage guarantee under exchangeability assumption for distribution-free cases. However, **prior knowledge** about the extent of the shift is necessary to maintain the coverage guarantee when a distribution shift happens. For instance, [R1] holds the coverage guarantee by constraining the test distribution within an $f$-divergence ball centered at a calibration distribution. Similarly, [R3] also develops the guarantee based on the knowledge of how distributions are contaminated.
>
> Furthermore, even if the theoretical coverage guarantee based on prior knowledge of distribution shift is obtained, it is **hard to keep prediction sets small** at the same time. For instance, the worst-case (WC) method [R1] holds the guarantee under distribution shift at the cost of inefficient large prediction sets.
>
> Experiments in the author rebuttal prove that our method can get **a better trade-off between the coverage and prediction interval size.**
>
> >Evaluating the difference under distribution shift using (5) doesn't seem very reasonable ...
>
> Eq. (5) characterizes the coverage difference under distribution shift. We minimize this coverage gap by Wasserstein distance regularization in Eq. (19) so we can use the quantile of calibration conformal scores to construct prediction sets.
>
> Specifically, we first use importance weighting [R2] with KDE to address covariate shift and approximate a weighted version of calibration conformal scores. Secondly, Wasserstein distance is reduced between test and the weighted calibration conformal score distributions to reduce the coverage gap caused by concept shift, so the coverage on test data approaches $1 –\alpha$.
>
> > it is quite strange why they replace this known value with $\hat{F}_{Q/P}(q^∗)$, generating an additional error.
>
> Appendix B justifies the error between $\hat{F} _{Q/P} (q^*)$ and  $\hat{F} _P (q)$ can be controlled by the increasing size of calibration data $n$. Secondly, $\hat{F} _{Q/P} (q^*)$ in Eq. (9) makes $D _{concept}$ a function of only one variable $q^*$, thus facilitating the introduction of NTW in the following sections. Please check Figure 1 (b) for the relationship between $\hat{F} _P (q)$ and $\hat{F} _{Q/P} (q^*)$ with given $\lceil{(1-\alpha)(n+1)} \rceil/n$.
>
> >  I'm a bit confused about equation (13), where again the exact equality seems to hold. The two empirical CDFs $\hat{F} _Q$ and $\hat{F} _{Q/P}$ are step functions where the jumps occur at the same values $v_1$,⋯,$v_n$, so it seems to me that exact equality holds.
>
> Eq. (13) does not hold exact equality. Because $\hat{F} _Q$ is the test conformal score CDF, it jumps over the test conformal scores $V_t$ as defined in Eq.(4), not calibration conformal scores $V_c$.
>
> >In that sense, the remaining discussion in section 3.2 sounds a bit odd. For example, the empirical CDFs have values exactly 1 for $v≥\max v_i$. What does the 'long tail' mean in this case?
>
> The purpose of truncation in Section 3.2 is to estimate Eq. (10) by modifying the Wasserstein distance in Eq. (13). In Eq. (10), each pair of $|\hat{F} _Q(v _i)-\hat{F} _{Q/P}(v _i)|$ has the same weight. In Eq. (13), the weight of $|\hat{F} _Q(v _i)-\hat{F} _{Q/P}(v _i)|$ is $v _{i+1}-v _i$, so we want $v _{i+1}-v _i$ is similar for different $i$ as well to approximate Eq. (10). However, the slop of conformal score CDFs tend to be flatter when the CDFs converge close to 1 but not reaching 1, which means $v _{i+1}-v _i$ there are quite large, so we need to truncate that part (long tail) in Eq. (15) to estimate Eq. (10). Table 1 justifies the necessity of truncation and normalization.
>
> > Could the authors clarify the motivation behind the need for 'multi-domain' inference?
>
> $P _{XY}$ is a mixture distribution of $Q _{XY}^{(e)}$ for $e\in\mathcal{E}$ as shown in Eq. (17). This can be a natural partition. For instance, $Q _{XY}^{(e)}$ can be the patient data of a hospital $e$, and $P _{XY}$ is the patient distribution of multiple hospitals. $1 – \alpha$ coverage on $P _{XY}$ can not ensure the coverage on  $Q _{XY}^{(e)}$, so we expect $1 − \alpha$ coverage on $Q _{XY}^{(e)}$ as well. Or $Q _{XY}^{(e)}$ can be traffic data from a single hour $e$, and $P _{XY}$ is the data collected within a day. Please check Appendix D for the experiment setups and dataset preprocessing details.
>
> > how do we know that the probability P((X,Y)∈e) is 1/M for every e in the partition?
>
> $P _{XY}$ is denoted as calibration domain and distribution at the same time. $Q _{XY}^{(e)}$ denotes the conditional distribution of $(x,y)$ given $(x,y)$ is from domain $e$.
>
> For the weight $1/M$, $P_{XY}$ can be the distribution of daily traffic sensor data, and $Q _{XY}^{(e)}$ is the data of an hour $e$, so the weight is $1/24$. Or taking the hospital example, we can fairly give $Q _{XY}^{(e)}$ of each hospital $e$ the same weight if their patient amounts are similar.
>
> **Reference**
>
> [R1] Cauchois M, Gupta S, Ali A, et al. Robust validation: Confident predictions even when distributions shift[J]. Journal of the American Statistical Association, 2024: 1-66.
>
> [R2] Tibshirani R J, Foygel Barber R, Candes E, et al. Conformal prediction under covariate shift[J]. Advances in neural information processing systems, 2019, 32.
>
> [R3] Sesia M, Wang Y X, Tong X. Adaptive conformal classification with noisy labels[J]. arXiv preprint arXiv:2309.05092, 2023.

---

> ### Comment · Reviewer_qkU5 · 2024-08-08
> **Follow-up discussion**
>
> Thank you for the authors' responses. However, I feel that many of the answers do not fully address the questions I raised. Below, I have included further questions and clarifications.
>
> 1. "The likelihood ratio is not assumed known and it is estimated by kernel density estimation (KDE)"
>
> This is exactly what I meant. In practice, the likelihood ratio must be estimated, even though the theory is often written as if it is known. For instance, the weighted conformal prediction method by Tibshirani et al. provides a comprehensive theoretical framework for the case where the likelihood ratio is known (and clearly states it), and then also mentions the need for estimation in practical scenarios. A similar level of clarity would be beneficial in this work (though I'm not very sure how much it will make sense in this work where there's no concrete theory even for the known-likelihood ratio setting).
>
>
> 2. "However, prior knowledge about the extent of the shift is necessary to maintain the coverage guarantee when a distribution shift happens..."
>
> I understand that distribution-free guarantees are not possible in the distribution-shift setting. My question is about the benefits of applying the conformal prediction framework in this context. Again as an example, Tibshirani et al. introduce weighted conformal prediction with a theoretical guarantee for the known likelihood ratio case, and Lei and Candes further develops this by showing that using an estimated likelihood ratio provides a coverage lower bound of $1-\alpha$-(TV distance based on the accuracy of estimation). This clearly illustrates the advantage of the conformal-type approach, as it requires only a good estimate of the likelihood ratio and offers 'distribution-free' inference with respect to the conditional distribution of $Y$ given $X$.
>
> In this work, the theoretical components are largely intuitive explanations rather than formal mathematical results. What I'm curious is why one should try to provide such intuition for the conformal-type approach. If some kind of distribution-free guarantee is not the goal, couldn't similar intuition be applied to other approaches that might perform better in practice?
>
> 3. Regarding equation (5):
>
> So it is true that $\hat{F}_P(q) = \lceil (1-\alpha)(n+1) \rceil/n$ holds exactly?
>
> I was asking whether it is necessary to replace this known value with one that introduces error, rather than the magnitude of the error. $D_\text{concept}$ is still a function of only $q^*$ if we simply plug in $\hat{F}_P(q) = \lceil (1-\alpha)(n+1) \rceil/n$, so the authors' response is not very convincing to me.
>
> ---Actually, upon closer inspection, it seems like the replacing term $\hat{F}_{Q/P}(q^*)$ is also exactly $\lceil (1-\alpha)(n+1) \rceil/n$? So it now seems to me that there is actually no replacement or error?
>
> 4. Thank you for the clarification regarding the difference between the supports of the two CDFs. As a quick follow-up, is this work more focused on large sample settings rather than small/finite sample settings? (since there are statements of the form "approximately holds when $n$ is large.") If so, could the authors further address point 1 above, concerning the need for a conformal approach whose main advantage is exact finite sample guarantees?
>
> 5. Regarding the mixture distribution representation, I'm not seeking examples. I am wondering, e.g., whether it is an assumption (that all mixture probabilities are equal, which seems quite strong), or something controllable through experimental design, etc. In the paper, it says the ``domain" is partitioned into $M$ bins, and then the mixture distribution representation suddenly appears. I'm just asking for some clarities here. The examples that the authors mentioned seem to be natural partitions rather than an experimental design, but then it doesn't seem very reasonable to just assume that all the mixture probabilities are equal.

---

> > ### Author Response · Authors · 2024-08-08
> >
> > Thank you for your comments.
> > >This is exactly what I meant. In practice, the likelihood ratio must be estimated, even though the theory is often written as if it is known. For instance, the weighted conformal prediction method by Tibshirani et al. provides a comprehensive theoretical framework for the case where the likelihood ratio is known (and clearly states it), and then also mentions the need for estimation in practical scenarios. A similar level of clarity would be beneficial in this work (though I'm not very sure how much it will make sense in this work where there's no concrete theory even for the known-likelihood ratio setting).
> >
> > Indeed, kernel density estimation (KDE) errors can be propagated to the CP coverage gap, and a theoretical analysis of the error propagation is beneficial. We would like to develop it in a later version.
> >
> > >In this work, the theoretical components are largely intuitive explanations rather than formal mathematical results. What I'm curious is why one should try to provide such intuition for the conformal-type approach. If some kind of distribution-free guarantee is not the goal, couldn't similar intuition be applied to other approaches that might perform better in practice?
> >
> > In Appendix C, we develop an upper bound guarantee for the empirical coverage gap, leveraging the fact that the coverage guarantee holds under the assumption of exchangeability, just like Section 1.2 of [R1].
> >
> > Specifically, Wasserstein distance is applied to quantify the extent of violating exchangeability and bound the empirical coverage gap with a probability related to the number of calibration and test samples.  This theoretical upper bound takes advantage of CP, while other uncertainty quantification methods, such as Bayesian neural networks, fail to do so.
> >
> > >Regarding equation (5)...
> >
> > The presentation in Section 3.1 should be improved and the logic should be as follows.
> >
> > First, let's consider the population case, thus $D_{joint}=F_Q(q)-F_P(q)$ by rewriting Eq. (5) in a population form.
> >
> > Similarly, $D_{covariate}$ is the reduced coverage gap after applying $q^*$ to $F_Q$.
> > $$
> > D_{covariate} = F_Q(q)-F_Q(q^*)
> > $$
> > Also, after importance weighting, calibration conformal scores are weighted, so the coverage on it is $F _{Q/P}(q^*)$. The remaining gap is caused by concept shift.
> > $$
> > D _{concept} =  F _{Q}(q^*)-F _{Q/P}(q^*)
> > $$
> > Because $F _{Q/P}(q^*)=F _{P}(q)=1-\alpha$ in population case, $D _{joint}=D _{covariate} +D _{concept} $ holds by
> > $$
> > D _{covariate} + D _{concept} =  F _Q(q)-F _Q(q^*) +  F _{Q}(q^*)-F _{Q/P}(q^*) = F_Q(q) - F _{Q/P}(q^*) =F_Q(q) - F _{P}(q)=D _{joint}.
> > $$
> > Now let's consider the empirical case. As you suggested, $\hat{F}_P(q)=\lceil(1-\alpha)(n+1)\rceil/n$. This is because the weight of each $v _i \in V _c$ is $1/n$, so there will be a conformal score just at the position $k$ satisfying
> >
> > $$
> > \sum_{i=1}^k \frac{1}{n}=\frac{\lceil(1-\alpha)(n+1)\rceil}{n}.
> > $$
> >
> > However, for $\hat{F}_{Q/P}(q^*)$, the weight of each $v _i \in V _c$  is $p_i$ after importance weighting, and we can not ensure that there is a position $k$ satisfying
> >
> > $$
> > \sum_{i=1}^k p_i=\frac{\lceil(1-\alpha)(n+1)\rceil}{n}.
> > $$
> >
> > As a result, $\hat{F} _{Q/P}(q^*) \geq \lceil(1-\alpha)(n+1)\rceil/n$ and the equality does not hold between $\hat{F} _{Q/P}(q^*)$ and $\hat{F}_P(q)$, so we need to bound the error in Appendix B due to discretization of CDFs.
> >
> > >Is this work more focused on large sample settings rather than small/finite sample settings?
> >
> > All works based on KDE rely on data accessibility as more samples will make KDE more accurate, but the work is not heavily dependent on large sample settings. For example, compared with other datasets, the airfoil dataset has the smallest sample size for each domain, 160, and the highest feature dimension, 5.  The performance of KDE and mRCP on it is acceptable in Figure 2 (a). Theoretical analysis of the estimation error is helpful.
> >
> > >whether it is an assumption (that all mixture probabilities are equal, which seems quite strong), or something controllable through experimental design, etc.
> >
> > This is not a requirement of the proposed method and it is just an experiment design, which means $P_{XY}$ does have to be divided equally.
> >
> > [R1] Barber R F, Candes E J, Ramdas A, et al. Conformal prediction beyond exchangeability[J]. The Annals of Statistics, 2023, 51(2): 816-845.

---

> ### Comment · Reviewer_qkU5 · 2024-08-08
>
> I appreciate the authors' detailed responses.
>
> I think I have now received answers regarding points 1, 2, and 4 in the follow-up discussion, and I hope there will be improvements and clarifications in the final version of the paper.
>
> I'd like to add some follow-up questions regarding points 3 and 5.
>
> 3. I see that for $\hat{F}_{Q/P}$, the exact equality does not hold---I think what I was trying to say is that it is still a known value that is close to $1−α$ (thanks for pointing this out). So, there is an error, and it now returns to the original question.
>
> In my original comment, what I meant was that viewing the empirical version of $F_P(q)$ as $\hat{F}_P(q)$
>
>  doesn't sound reasonable, since $\hat{F}_P$
>
>  and $q$
>  are derived from the same datasets (which involves double-dipping). It also applies to $\hat{F}_{Q/P}(q^*)$.
>
> The main question that remains unanswered is whether, if we proceed with this empirical version, it is necessary to replace this known value with something that includes an error. Are there any challenges in applying a similar approach, if we just plug in the exact value?
>
> 5. I understand that it might not make much sense to ask if this statement is some kind of assumption, as it is not as though a theorem is based on it. However, I think the following clarifications would be helpful:
>
> - Is the idea of the proposed method indeed based on model (17), which suggests that the sampling distribution is a mixture distribution with equal mixture probabilities?
>
> - If it indeed requires equal mixture probabilities, then 'partitioning the domain into $M$ bins' is not an accurate statement. Is this what the authors agree?
>
> - My concern is that the dataset examples still seem more fitting to a setting where the domain is partitioned according to some natural rule, rather than by experimental design. So I believe some justification is needed here.

---

> > ### Author Response · Authors · 2024-08-09
> >
> > Thank you for your questions. We hope the explanation below is helpful.
> > >In my original comment, what I meant was that viewing the empirical version of $F_P(q)$ as $\hat{F}_P(q)$ doesn't sound reasonable, since ...
> >
> > We think your point is: since $q$ is actually calculated from $\hat{F} _P$ according to Eq.(1), it looks strange to apply $q$ backward to $\hat{F} _P$ and calculate the coverage on calibration data as $\hat{F} _P(q)$.
> >
> > Indeed the coverage on calibration data can be written as $\lceil(1+\alpha)(n+1)\rceil / n$, so Eq.(8) can be written as
> >
> > $$
> > \hat{F}_Q(q^*)-\lceil(1+\alpha)(n+1)\rceil / n.
> > $$
> >
> > So why don't we optimize based on this form?
> >
> > Based on the definition of  $D _{concept}$, coverage on importance-weighted calibration data is $\hat{F} _{Q/P}(q^*)$ instead of  $\lceil(1+\alpha)(n+1)\rceil /n$, so $D _{concept} =\hat{F} _{Q}(q^*)-\hat{F} _{Q/P}(q^*)$ .
> >
> > In other words, the position of the discretization error, $\epsilon$, should be
> >
> > $$
> > D_{joint}=D_{covariate}+D_{concept}+ \epsilon
> > $$
> >
> > If we optimize $D _{concept} =\hat{F} _{Q}(q^*)-\lceil(1+\alpha)(n+1)\rceil / n$, we are actually blaming concept shift for the introduction of $ \epsilon$. However,  $\epsilon$ is caused by discretization instead of concept shift.
> >
> > Potential risks of differentiability may be raised if optimizing $\hat{F} _{Q}(q^*)-\lceil(1+\alpha)(n+1)\rceil / n$ as well.
> >
> > >I understand that it might not make much sense to ask if this statement is some kind of assumption, as it is not as though a theorem is based on it. However, I think the following clarifications would be helpful:
> >
> > >Is the idea of the proposed method indeed based on model (17), which suggests that the sampling distribution is a mixture distribution with equal mixture probabilities?
> >
> > The idea of the proposed method is not based or limited on Eq.(17), but we think coverage on subdomains is a practical application scenario of the proposed method. Eq.(17) represents an equally weighted special case.
> >
> > >If it indeed requires equal mixture probabilities, then 'partitioning the domain into M bins' is not an accurate statement. Is this what the authors agree?
> >
> > Also, even for equally weighted mixture probabilities, we do not find statements like 'partitioning the domain into $M$ bins' in our work and this statement is not accurate. $M$ bins sound like the subdomains have disjoint feature spaces, but this is not the case for Eq. (17). Line 155 may mislead to 'partitioning into bins' and should be improved.
> >
> > >My concern is that the dataset examples still seem more fitting to a setting where the domain is partitioned according to some natural rule, rather than by experimental design. So I believe some justification is needed here.
> >
> > Indeed, Eq.(17) can be improved to a more general form, like $P_{XY}$ is a convex combination of $Q_{XY}^{(e)}$ for $e\in\mathcal{E}$, without limiting the weight of each $Q_{XY}^{(e)}$ as $1/M$. Then we should clarify the weights when we mention experiments on specific datasets. This will make the method more generally applicable.

---

> > > ### Comment · Reviewer_qkU5 · 2024-08-09
> > >
> > > This will probably be my last comment.
> > >
> > > I appreciate the authors' answers throughout this long discussion. However, I have to say that I don't feel the answers are very consistent and that they don't completely address the raised questions.
> > >
> > > Regarding the "coverage gap" argument in Section 3.1, I believe we need to start from the fundamental ideas behind it to clarify everything from the ground up and ensure we're on the same page.
> > >
> > > As stated in the authors' previous answers, it's clearer in terms of intuition to write everything in terms of population CDFs, with replacing them by sample versions being a separate step that comes later.
> > >
> > > So let's look at $D_{joint} = F_Q(q) - F_P(q)$. In the paper, the authors use the term 'coverage difference' under joint distribution shift to denote $D_{joint}$. Here, to use the term 'coverage', the first thing to do is to clearly state whose coverage it is. For $D_{joint}$, it is probably the standard split conformal prediction set, since it is where $q$ comes from. Therefore,
> > >
> > > $D_{joint}$ in eq (5) represents: (coverage of split conformal under distribution shift) - (coverage of split conformal without distribution shift)
> > >
> > > Now we can clarify other terms similarly.
> > >
> > > $D_{covaraite}$ in eq (7) represents: (coverage of split conformal under distribution shift) - (coverage of weighted conformal under distribution shift).
> > >
> > > $D_{concept}$ in eq (8) represents: (coverage of weighted conformal under distribution shift) - (coverage of split conformal without distribution shift)
> > >
> > > So how can $D_{covariate}$ and $D_{concept}$ be termed as 'coverage differences' when they are not about a coverage of one prediction set?
> > >
> > > For example, in the paper, the authors describe $D_{concept}$ as "coverage difference caused by concept shift", but this is quite vague and doesn't seem like a well defined statement. Whose coverage is it referring to?
> > >
> > > After all these steps, we need to plug in empirical versions of the CDFs that appear, and what I asked earlier but did not get an answer to is that there is a clear double-dipping issue, which makes the estimation unreliable.
> > >
> > > By the way, to clarify, I'm not trying to say that the methodology the authors provide doesn't seem reasonable. I believe the method itself has some value. What I'm trying to convey is that the 'theoretical arguments' provided before the methodology are difficult to follow and need significant clarification and improvement.
> > >
> > > Similarly, the most recent answer from the authors is also hard to follow. Where does the 'discretization' come from, and what does it mean in this context? What do the authors mean by 'discretization error' $\epsilon$? What do the authors mean by 'blaming concept shift'?
> > >
> > > Could the authors provide clear explanations on all the questions raised above?
> > >
> > >
> > > Regarding the mixture distribution representation, the original issue was that the paper states that the domain is decomposed into $M$ bins, and then the mixture distribution representation suddenly appears. I was just asking for clarification on this. In their previous answers, the authors said that "it is just an experiment design", but it is something that appeared in Section 4 where the method is introduced, not in Section 5 where the experimental results are provided. So it sounded like the theoretical setting assumes the practitioner determines the mixture probability and draws data based on it. However, based on the examples provided by the authors and their most recent answer, this doesn't seem to be the case, and I can't quite get what the authors are trying to say.
> > >
> > > The authors mentioned traffic and hospital data as examples and referred to decomposing the domain based on hours, suggesting a partition of the domain. In fact, they stated: " $P_{XY}$ is a mixture distribution of $Q_{XY}^{(e)}$  for $e \in \mathcal{E}$ as shown in Eq. (17). This can be a natural partition". However, in the most recent answer, they are now saying that it is not necessarily a partition and there can be overlaps.

---

> > > > ### Author Response · Authors · 2024-08-11
> > > >
> > > > Thank you for your comments and questions helping us find out the problems of the work.
> > > >
> > > > For your concerns about Section 3.1, the theory should be more organized and logical.
> > > >
> > > > First of all, we should introduce $D_{covariate}$ and $D_{concept}$ **as we discussed in our first comment**. And then derive Eq.(5) why $D_{covariate}+D_{concept}=D_{joint}$ in population case.
> > > >
> > > > Based on the population case, the empirical case can be developed by analyzing how large the error between $\sum_{i=1}^k p_i$ and $\lceil(1+\alpha)(n+1)\rceil/n$ can be, with $v_k=q^*$. This error is caused because of the finite number of samples (discretization of continuous CDFs).
> > > >
> > > > In other words, Eq.(8) is the derivation of $D_{covariate}+D_{concept}=D_{joint}$ under the population case, but now it misleads the readers to take it as the definition of $D_{concept}$.
> > > >
> > > > >Regarding the mixture distribution representation, the original issue was that the paper states that the domain is decomposed in M bins...
> > > >
> > > > In our most recent comment, we just mean Algorithm 1 is also applicable if rewrite Eq. (17) to a more general form, a convex combination of Q^{(e)}_{XY}. Eq. (17) is a special case of convex combinations.
> > > >
> > > > Experiments in Section 5 can be redesigned to validate when subdomains have equal and unequal weights.
> > > >
> > > > >However, in the most recent answer, they are now saying that it is not necessarily a partition and there can be overlaps.
> > > >
> > > > We would like to clarify the meaning of "partition" and "overlaps".
> > > > A domain can be decomposed into subdomains as Eq. (17). The decomposition (or partition) means an element can not be a member of multiple groups at the same time. For instance, a person can not be a patient of two hospitals.
> > > > 'Overlaps' means subdomains can have overlapped (not disjoint) space (feature space and target space). For example, two hospitals can have patients at 40 years old, if we take age as a feature.

---

### Official Review · Reviewer_V1Vs · 2024-07-21

**Soundness:** 2
**Presentation:** 2
**Contribution:** 2
**Rating:** 4
**Confidence:** 2

**Summary:**

The authors propose a method to train a predictive model (a regressor in their experiments) that minimizes an objective comprising the average performance loss across multiple domains (environments) along with a penalty term for the normalized truncated Wasserstein (NTW) distance between the non-conformity score CDFs of each environment and the importance-weighted one used to address covariate shift. Their experimental results demonstrate that the proposed NTW distance objective is correlated with coverage differences due to concept shift and can achieve different tradeoffs with the average prediction residuals, thereby reducing this gap.

**Strengths:**

Overall, the paper is easy to follow and the reasoning behind the proposed formulation is compelling. The problem that the authors address is relevant. The experimental results show that the proposed NTW distance is capturing the coverage difference due to concept shift.

**Weaknesses:**

While the motivation behind the regularization is compelling, it is not entirely clear, both empirically and theoretically, what specific benefits it offers over other state-of-the-art approaches that address differences in the non-conformity score distributions. To highlight the advantages of the proposed NTW regularization over simply minimizing the prediction residuals and then applying various post-hoc conformal prediction techniques, I suggest that the authors demonstrate the validity and efficiency of the prediction intervals obtained for different alphas (error levels) on test data. Additionally, they should provide empirical comparisons or some fundamental discussion/theoretical results in relation to the following approaches:

* Split conformal prediction and group conditional split conformal prediction (where each environment is treated as a separate group). The latter can include standard SCP conditioned on each group or an approach such as the one in section 4.1 of [Barber et al. 2020, "The Limits of Distribution-Free Conditional Predictive Inference"] or BatchGCP/BatchMVP in [Jung et al. 2022, "Batch Multivalid Conformal Prediction"].

* Performance of the covariate shift split conformal prediction, as already discussed in the paper. For example, if this is built on top of a model that minimizes ERM, DRO or V-REx does the proposed approach provide better prediction sets in terms of conditional coverage/validity on the domains.

* An adaptive approach such as the one by [Amoukou and Brunel 2023, Adaptive Conformal Prediction by Reweighing Nonconformity Score].

Providing such comparative results or analysis would significantly strengthen the paper's argument.

I also think the authors should discuss how their work relates to [Barber, Rina Foygel, et al. "Conformal prediction beyond exchangeability"], where it is suggested that the non-conformity scores should be weighted based on the total variation distance between the source and target distributions. This approach could potentially serve as another baseline to consider, given the distance between the distributions of the non-conformity scores under P and Q(e).

**Questions:**

In the experiments, how do the authors compute the importance weights for covariate shift?

How do you pick \sigma?

Revise typos such as "Dcoviriate" in Figure 1 caption.

**Limitations:**

yes, they mention some of the limitations of the proposed work.

---

> ### Author Rebuttal · Authors · 2024-08-07
>
> Thank you for your questions.
> > What specific benefits it offers over other state-of-the-art approaches that address differences in the non-conformity score distributions? I suggest that the authors demonstrate the validity and efficiency of the prediction intervals obtained for different alphas on test data.
>
> When a theoretical coverage guarantee under distribution shift is obtained, it is hard to keep prediction sets small at the same time. For instance, as mentioned in lines 41-43 the worst-case (WC) method [R1] holds the guarantee under distribution shift at the cost of inefficient large prediction sets by applying the highest $\lceil{(1-\alpha)(n+1)} \rceil/n$ quantile to all test distributions.
>
> The proposed method can get **a good trade-off between the coverage and prediction interval size** because of two reasons. First, we do not follow conservative methods as mentioned above. By minimizing the distribution discrepancy via NTW, we make the calibration and test distribution score distributions satisfy exchangeability better, so we can use the calibration conformal score quantile (not the largest quantile of multiple test distributions) to generate prediction sets. Secondly, prediction interval size is related to the magnitude of conformal scores which is absolute residual for regression tasks, and Eq. (19) can balance the residual loss and NTW loss.
>
> We demonstrate the efficiency of the proposed method in author rebuttal.
>
> >Split conformal prediction and group conditional split conformal prediction ... The latter can include ...  section 4.1 of [Barber et al. 2020, "The Limits of Distribution-Free Conditional Predictive Inference"] or BatchGCP/BatchMVP in [Jung et al. 2022, "Batch Multivalid Conformal Prediction"].
>
> The performance of Split CP (SCP) with a model trained by ERM is highlighted in the author rebuttal.
>
> For the comparison with BatchMVP [R2], first BatchMVP does not decompose distribution shifts between each group into covariate and concept shifts. Secondly, Batch MVP is trained based on a given $\alpha$ value, so the multi-valid coverage guarantee only holds at a specific error level. However, mRCP minimizes the distribution discrepancy between calibration and test conformal scores, so a single trained model can make the coverage on test data approach $1-\alpha$ no matter how $\alpha$ value changes.
>
> For the comparison with Section 4.1 of [R3], in Eq. (10) of [R3], it takes the supremum of quantiles for groups in $\hat{\mathfrak{X}} _{n _1}$ to ensure the coverage for any group $\mathcal{X}\in\hat{\mathfrak{X}} _{n _1}$. As a result, [R3] has the same problem as [R1] and is likely to cause unnecessarily large prediction intervals.
>
> >Performance of the covariate shift split conformal prediction...
>
> The result of importance-weighted SCP [R6] with a model trained by ERM is highlighted in author rebuttal.
>
> Models trained by V-REx can be applied to standard SCP or importance-weighted SCP. The result of standard SCP based on V-REx is shown as the blue curves in Figure 2. The result of importance-weighted SCP based on V-REx will be worse than standard SCP. Because V-REx aligns the unweighted score distributions by minimizing its regularization term, weighting conformal scores afterward will disturb the previous alignment and make score distributions more discrepant, thus enlarging the coverage gap.
>
> The same also happens to the importance-weighted SCP with the model trained by DRO.
>
> >An adaptive approach such as the one by [Amoukou and Brunel 2023, Adaptive Conformal Prediction by Reweighing Nonconformity Score]
>
> [R4] applies a novel conformal score function to make the prediction set more customized to the input $x$. However, it is based on the exchangeability assumption, having a different problem setup from ours. The position of [R4] is in the first row of Table 2.
>
> >the authors should discuss how their work relates to [Barber, Rina Foygel, et al. "Conformal prediction beyond exchangeability"]...
>
> [R5] uses total variation to bound coverage gap. However, total variation measures half the absolute area between two probability density functions, without considering how the probabilities are distributed across the support of the distributions. On the contrary, Wasserstein distance considers how probabilities are dispersed along the support. Therefore, the same total variations can come up with different Wasserstein distances in Figure 1 (c). Wasserstein distance can reflect the overall difference of two conformal score CDFs along the support, but total variation fails to do so.
>
> > How do you pick $\sigma$
>
> The experiment results can be different with different $\sigma$ values.  $\sigma$ is selected based on the properties of conformal score distributions. Usually, $\sigma$ should be smaller (truncate more) for more concentrated distributions. We choose $\sigma$=0.8 for MLP and 0.95 for physics-informed models, because MLP has a better fitting ability, making conformal scores more concentrated.
>
> **Reference**
>
> [R1] Cauchois M, Gupta S, Ali A, et al. Robust validation: Confident predictions even when distributions shift[J]. Journal of the American Statistical Association, 2024: 1-66.
>
> [R2] Jung C, Noarov G, Ramalingam R, et al. Batch multivalid conformal prediction[J]. arXiv preprint arXiv:2209.15145, 2022.
>
> [R3] Foygel Barber R, Candes E J, Ramdas A, et al. The limits of distribution-free conditional predictive inference[J]. Information and Inference: A Journal of the IMA, 2021, 10(2): 455-482.
>
> [R4] Amoukou S I, Brunel N J B. Adaptive conformal prediction by reweighting nonconformity score[J]. arXiv preprint arXiv:2303.12695, 2023.
>
> [R5] Barber R F, Candes E J, Ramdas A, et al. Conformal prediction beyond exchangeability[J]. The Annals of Statistics, 2023, 51(2): 816-845.
>
> [R6] Tibshirani R J, Foygel Barber R, Candes E, et al. Conformal prediction under covariate shift[J]. Advances in neural information processing systems, 2019, 32.

---

### Official Review · Reviewer_Ab2r · 2024-07-23

**Soundness:** 1
**Presentation:** 1
**Contribution:** 2
**Rating:** 3
**Confidence:** 4

**Summary:**

The paper tackles the challenge of obtaining conformal predictions that remain robust under distribution shifts. This is an important issue in many machine learning applications where the underlying data distribution may change across the training (or calibration) and test data sets. The authors propose an algorithm that appears promising based on empirical results. However, significant improvements are needed in terms of writing quality, clarity, citation of relevant literature, mathematical rigor, and explanation of the main ideas. Addressing these substantial weaknesses would make the paper more accessible and impactful for the research community.

**Strengths:**

- The problem of robust conformal predictions under distribution shifts is timely and relevant.

- The paper introduces a concrete algorithm that demonstrates promising performance in some practical scenarios.

**Weaknesses:**

- Writing Quality: The paper is difficult to read and understand, even for experts. Key concepts are not clearly explained, and the text contains numerous awkward or unclear sentences, as well as pervasive grammatical errors and typos. Some sections seem poorly written, possibly by AI, while others could have been significantly improved with better editing.

 - Missing References: Important related works, such as "Conformal prediction beyond exchangeability" by Barber et al. (2023), are not discussed, which limits the paper's contextual grounding in existing literature.

- Lack of Statistical/Mathematical Rigor: The mathematical details in the paper are imprecise. Assumptions and approximations are not clearly stated, and there is a frequent confusion between population and sample quantities in key sections.

 - Unclear Core Idea: The main idea of the proposed algorithm, particularly how it handles concept shift through covariate shift adjustments (as in Equation 10), is not clearly articulated and remains confusing.

**Questions:**

- Writing Quality: The current writing is difficult to understand, and professional editing may be an option if needed. Parts of the paper read like they were written by Chat-GPT, but not in a good way, with lots of very ackwards sentences and passages that don't make sense. Other parts could have been improved by Chat-GPT, due to the large number of grammatical errors and typos.

- Clarity of Introduction: The introduction does not clearly explain the problem being addressed or the novelty of the proposed approach. Could you clarify these points to help readers understand the significance of your work?

- Mathematical Precision: The paper seems to systematically confuse population and sample quantities, starting from Equations (7)-(9) and (10). For instance, why is an empirical distribution used in Equation (7) instead of a population distribution, in the definition of a quantity (the coverage gap) which should be a population quantity? Additionally, the "expected value" in Equation (10) does not seem correctly formulated, since the left-hand-side of the equation is a population quantity (hence fixed) but the right-hand-side is a sample quantity (hence random). I don't think this is correct math.

- Assumptions and Justifications: Lines 112-115 introduce an assumption that is neither explained nor motivated, followed by a vague statement about a small error bound. Can you provide a clearer explanation and justification for this assumption? Is this an assumption or an approximation?

- Core Concept Explanation: The explanation of handling concept shift via covariate shift, particularly in Equation (10), is confusing. Could you clarify this core idea, as it is central to your paper's contribution? It seems from Equation (10) that concept shift was reduced to a (much simpler) covariate shift problem. I don't understand how this happened.

- Background on Conformal Prediction: Section 2.1 lacks precision. For example, Equation (2) describes a special case for regression, but the referenced works use different approaches. Can you clarify these points to make the section more accessible to a broad readership? Is this paper focusing on regression or classification? Is it limited to a specific type of conformity scores, such as that in Equation (2), or is it more general?

- Connection to Related Work: Is there a connection between the coverage difference you considered and that studied in "Adaptive conformal classification with noisy labels" by Sesia et al. (2023)? It seems there might be a relationship, even though Sesia et al. consider a specific case of distribution shift. Can you discuss any similarities or differences?

**Limitations:**

The paper's main limitations are its poor writing quality and lack of mathematical precision. These issues make it difficult to understand the main ideas and verify the soundness of the proposed method.

---

> ### Author Rebuttal · Authors · 2024-08-07
>
> Thank you for your suggestions and questions.
> >Missing References: Important related works, such as "Conformal prediction beyond exchangeability"...
>
> [R1] uses total variation to bound coverage gap. However, total variation measures half the absolute area between two probability density functions without considering how the probabilities are distributed across the support of the distributions. On the contrary, Wasserstein distance considers how probabilities are dispersed along the support. Therefore, the same total variations can come up with different Wasserstein distances in Figure 1 (c). As a result, Wasserstein distance can reflect the overall difference of two conformal score CDFs along the support, which is related to the coverage gap, but total variation fails to do so.
>
> >Why is an empirical distribution used in Equation (7) instead of a population distribution, in the definition of a quantity (the coverage gap) which should be a population quantity?
>
> We focus on the empirical coverage gap with finite samples. This is widely discussed and important to practical applications, such as Section 1.2 of [R1]. **The empirical coverage gap can be bounded with a probability related to the numbers of calibration and test samples** in Eq. (30) of Appendix C. Besides, the upper bound builds a connection between population forms of conformal score distributions and the empirical coverage gap.
>
> >Lines 112-115 introduce an assumption that is neither explained nor motivated...
>
> The assumption in lines 112-115 can be presented more rigorously. In Appendix B, we develop the following inequality for Eq. (7).
>
> $$
> D_{concept}=\hat{F}_Q(q^{\ast})-\hat{F}_P(q)=\hat{F}_Q(q^{\ast})-\hat{F} _{Q/P}(q^{\ast})+\hat{F} _{Q/P}(q^{\ast})-\hat{F}_P(q) \leq \hat{F}_Q(q^{\ast})-\hat{F} _{Q/P}(q^{\ast})+|\hat{F} _{Q/P}(q^{\ast})-\hat{F}_P(q)|.
> $$
>
> Since the error can be bounded in Appendix B as follows and can be controlled by increasing the size of calibration data $n$,
>
> $$
> |\hat{F} _{Q/P}(q^{\ast})-\hat{F}_P(q)| < \max(\hat{F} _{Q/P}(q^{\ast} _{+})-\lceil{(1-\alpha)(n+1)} \rceil/n, \hat{F} _{P}(q _{+})-\lceil{(1-\alpha)(n+1)} \rceil/n ),
> $$
>
> we can focus on $\hat{F}_Q(q^{\ast})-\hat{F} _{Q/P}(q^{\ast})$.
>
> Intuitively, since $q$ and $q^*$ are $\lceil{(1-\alpha)(n+1)} \rceil/n$ quantiles of $\hat{F} _{P}$ and $\hat{F} _{Q/P}$ respectively, with a large $n$, it is reasonable to expect the difference between $\hat{F} _{Q/P}(q^{\ast})$ and $\hat{F}_P(q)$ is very small. Please check Figure 1 (b) for the relationship between $\hat{F} _{Q/P}(q^{\ast})$ and $\hat{F}_P(q)$ with a given $\lceil{(1-\alpha)(n+1)} \rceil/n$ value.
>
> >The explanation of handling concept shift via covariate shift, particularly in Equation (10), is confusing...
>
> **We do not intend to address concept shift by covariate shift in Eq. (10)**. At first, we approximately address covariate shift by importance weighting [R4] based on kernel density estimation. Therefore, we can estimate the weighted calibration conformal score CDF $\hat{F} _{Q/P}$ from the unweighted one $\hat{F} _P$.  The remaining gap between $\hat{F} _{Q/P}$ and the test conformal score CDF $\hat{F} _{Q}$ is caused by concept shift in Eq. (8). Then, we quantify the distribution discrepancy between $\hat{F} _{Q/P}$ and $\hat{F} _{Q}$ by NTW, and minimize it during training.
>
> mRCP can make the coverage on test data approach $1-\alpha$, rather than applying conservative methods, like the worst-case (WC) method [R5] as mentioned in lines 41-43, which is likely to cause overestimated coverage and prediction sets.
>
> We demonstrate **a better trade-off between coverage and prediction intervals** of the proposed method compared with WC method in the author rebuttal.
>
> > Section 2.1 lacks precision. For example, Equation (2) describes a special case for regression...Is it limited to a specific type of conformity scores, such as that in Equation (2), or is it more general?
>
> We focus on regression tasks as mentioned in line 65 with the conformal score function defined in Eq. (2), which is **commonly used** for regression tasks [R1][R2][R4]. The proposed method is also applicable to other conformal score functions for regression tasks, like the one proposed in localized split CP [R6].
>
> > Is there a connection between the coverage difference you considered and that studied in "Adaptive conformal classification with noisy labels" by Sesia et al. (2023)?
>
> [R3] considers CP on classification problems whereas we focus on CP on regression tasks. [R3] aims to maintain the coverage guarantee if the labels of calibration samples are contaminated during sampling, which means the exchangeability assumption is violated if test samples are drawn from the same population distribution. Therefore, [R3] considers concept shift but not covariate shift as the marginal distribution of features does not change.
>
> Our work investigates the coverage difference under joint distribution shift between calibration and test distributions, which means the covariate shift and concept shift can occur simultaneously.
>
>
> **Reference**
>
> [R1] Barber R F, Candes E J, Ramdas A, et al. Conformal prediction beyond exchangeability[J]. The Annals of Statistics, 2023, 51(2): 816-845.
>
> [R2] Romano Y, Patterson E, Candes E. Conformalized quantile regression[J]. Advances in neural information processing systems, 2019, 32.
>
> [R3] Sesia M, Wang Y X, Tong X. Adaptive conformal classification with noisy labels[J]. arXiv preprint arXiv:2309.05092, 2023.
>
> [R4] Tibshirani R J, Foygel Barber R, Candes E, et al. Conformal prediction under covariate shift[J]. Advances in neural information processing systems, 2019, 32.
>
> [R5] Cauchois M, Gupta S, Ali A, et al. Robust validation: Confident predictions even when distributions shift[J]. Journal of the American Statistical Association, 2024: 1-66.
>
> [R6] Han X, Tang Z, Ghosh J, et al. Split localized conformal prediction[J]. arXiv preprint arXiv:2206.13092, 2022.

---

> > ### Comment · Reviewer_Ab2r · 2024-08-09
> >
> > Thank you for your understanding of my feedback. However, I believe that my concerns go beyond what can be resolved through this discussion alone. I believe this paper requires significant improvements in both the clarity of writing and the level of mathematical rigor before it can undergo another thorough review.
> >
> > Other reviewers have also expressed concerns regarding the clarity of the paper, the theoretical foundations and relation to other approaches, and the soundness of the mathematical arguments. While the method appears to perform well in certain empirical scenarios, it's not enough. It either needs to be clearly presented as a paper suggesting a heuristic algorithm, or the theoretical ideas behind it need to be articulated much better.

---

### Author Rebuttal · Authors · 2024-08-07

**Kernel Density Estimation for Likelihood Ratio**

The likelihood ratio is not assumed to be known and is approximated by kernel density estimation (KDE), which can estimate the calibration and test feature distributions. In our experiments, we applied the Gaussian kernel, a positive function of  $x\in\mathbb{R}^d$ as follows, where ‖∙‖ is Euclidean distance and $h$ is bandwidth.
$$
K(x,h)=\frac{1}{(\sqrt{2\pi}h)^d} e^{\frac{-‖x‖^2}{2h^2}}
$$
Given this kernel form, the density estimated at a position $x_p$ within a group of points $x_{1:n}$ is given by
$$
ρ_K (x_p )=\sum_{i=1}^n K(x_p-x_i;h).
$$
To find the optimized bandwidth value for each dataset, we applied scikit-learn package [R1] using the grid search method with a bandwidth pool. With the approximated calibration and test feature distribution, we can calculate the likelihood ratio to implement importance weighting.

**Experiment Results of Coverage and Prediction Interval**

As mentioned in lines 41-43, the worst-case (WC) method [R2] only selects the highest $\lceil{(1-\alpha)(n+1)} \rceil/n$ quantile for all test distributions. Even if the theoretical coverage guarantee is ensured, it will cause excessively large prediction sets and overestimated coverage for test distributions with smaller quantiles.

We provide experimental results about coverage and prediction intervals by mRCP in comparison with the worst-case (WC) method. We denote $C^{(e)}$ and $I^{(e)}$ as the coverage and prediction interval length of domain $e$ with a given $1-\alpha$. $\mathbb{E}_e [C^{(e)}]$ and $\mathbb{E}_e [I^{(e)}]$ are expectations of $C^{(e)}$ and $I^{(e)}$ over $e\in \mathcal{E}$.

In **Figure 5** of the attachment, even if mRCP coverage is not guaranteed at least $1-\alpha$, it approaches $1-\alpha$ very close with relatively small standard deviations. However, WC causes excessive coverage. As a result, WC generates larger prediction intervals compared with mRCP in **Figure 6**, causing less prediction efficiency.

$\beta$ value of mRCP is selected as 1 for the airfoil dataset and 10 for the other six datasets. The same MLP architectures are applied to WC and mRCP. Other experiment settings are the same as in Appendix D.

**Spearman’s Rank Coefficient of the Experiment Results in Section 5.2**

We provide the experiment result of Spearman’s rank coefficient in **Table 6** in the attachment as supplementary material to Section 5.2. In Table 6, NTW holds the highest Spearman coefficient on average, indicating a strong positive correlation with Eq. (10).

**Explicitly Highlighted Baselines in Figure 2**

The performance of standard Split CP (SCP) with a model trained by V-REx is shown in Figure 2 as the blue curves. Therefore, the result of standard SCP with empirical risk minimization (ERM) is presented as the most left side of blue curves (V-REx) in Figure 2, where the regularization weight $\beta$ of V-REx is small, and thus V-REx can be regarded as ERM.

For importance-weighted SCP [R3] with a model trained by ERM, its performance is presented as the most left side of the orange curve (mRCP) of Figure 2.  Because at the left side of the orange curve (mRCP), the weight $\beta$ of NTW regularization is small enough that mRCP can be regarded as ERM. How wide coverage gap can be reduced by importance weighting depends on the extent of covariate shift between test and calibration distributions

We take **Figure 2 (a)** as an example in the attachment to highlight the results of these two setups. The results for other datasets can be checked in other subplots of Figure 2 in the same way.

**Reference**

[R1] Pedregosa F, Varoquaux G, Gramfort A, et al. Scikit-learn: Machine learning in Python[J]. the Journal of machine Learning research, 2011, 12: 2825-2830.

[R2] Cauchois M, Gupta S, Ali A, et al. Robust validation: Confident predictions even when distributions shift[J]. Journal of the American Statistical Association, 2024: 1-66.

[R3] Tibshirani R J, Foygel Barber R, Candes E, et al. Conformal prediction under covariate shift[J]. Advances in neural information processing systems, 2019, 32.

---

### Decision · Program_Chairs · 2024-09-25

**Decision:**

Reject

**Comment:**

The paper considers the problem of construction of conformal prediction sets that remain robust under distribution shifts. The authors propose a method to train a predictive model that minimizes an objective comprising the average performance loss across multiple domains along with a penalty term for the normalized truncated Wasserstein distance between the non-conformity score CDFs of each environment and the importance-weighted one used to address covariate shift.

While the reviewers found the proposed algorithm to be novel, and the problem setting to be important, they suggested that the paper needs to be improved in terms of writing quality, clarity, citation of relevant literature, mathematical rigor, and explanation of the main ideas. I concur with this evaluation and recommend a rejection of the paper in its current form. I encourage the authors to revise and resubmit: once these issues are addressed, the paper will be an impactful addition to the CP community.